# GraphChef: Decision-Tree Recipes to Explain Graph Neural Networks

**Peter Müller, Lukas Faber, Karolis Martinkus, Roger Wattenhofer**
ETH Zurich, Switzerland
`{lfaber,mkarolis,wattenhofer}@ethz.ch`

## Abstract

We propose a new self-explainable Graph Neural Network (GNN) model: GraphChef. GraphChef integrates decision trees into the GNN message passing framework. Given a dataset, GraphChef returns a set of rules (a recipe) that explains each class in the dataset unlike existing GNNs and explanation methods that reason on individual graphs. Thanks to the decision trees, the GraphChef recipes are human-comprehensible. We also present a new pruning method to produce small and easy-to-digest trees. Experiments demonstrate that GraphChef reaches comparable accuracy to non-self-explainable GNNs, and the produced decision trees are indeed small. We further validate the correctness of the discovered recipes on datasets where explanation ground truth is available: Reddit-Binary, MUTAG, BA-2Motifs, BA-Shapes, Tree-Cycle, and Tree-Grid.

## 1 Introduction

Graphs abstractly represent complex relational data in a myriad of applications and play a crucial role in, e.g., chemistry, engineering, social sciences, or transportation. Graph Neural Networks (GNNs) are a popular but black-box machine learning model for graph-based domains. GNNs classify individual graphs, as in "Is this protein (represented as a graph) an enzyme?" Naturally, explaining how these black-box GNNs make their predictions is also important. Most existing works aim to explain GNN predictions by identifying key nodes (Figure 1a) and edges (Figure 1b) or by finding similar examples or subgraphs (Figure 1c). Each of these existing methods can highlight the importance of the double "Sheet" motive, but no method helps us to understand why it is important. In this paper, we want to drive explanation further and 1) understand not only which inputs are important, but also how they are used and 2) understand the dataset as a whole as to really answer the question "What makes a protein an enzyme?". GraphChef builds a recipe in the form of decision trees (Figures 1d+1e). The recipe shows hows Sheets contribute to the Enzyme class. The PROTEINS dataset shown is an easy example dataset without intermediate layers. In further experiments, we show that GraphChef works equally well for various explanation benchmarks that require graph reasoning. In summary, our contributions are as follows:

- While traditional GNNs are based on synchronous message passing (Loukas, 2020), we propose a new layer that is inspired by a simplified distributed computing model known as the stone-age model (Emek and Wattenhofer, 2013). In this model, the nodes use a small categorical space for states and messages. The stone-age model is simple and as such suitable for interpretation while retaining a high theoretical expressiveness. We call our new layer *dish* (DIfferentiable Stone-"H").

- We distill the multi-layer perceptrons in all dish layers to decision trees. We call the resulting model GraphChef. GraphChef abstractly expresses the reasoning of the dish GNN with a series of decision trees (the recipe). See Figure 1 for a complete example.

- We propose a way to collectively prune the decision trees in GraphChef. Pruning may affect accuracy, but also gives simpler explanations. GraphChef hence allows for a trade-off between accuracy and simplicity.

- We introduce an importance propagation scheme through the recipes to allow GraphChef to compute node-level importance scores similar to orthodox GNN explanation methods.

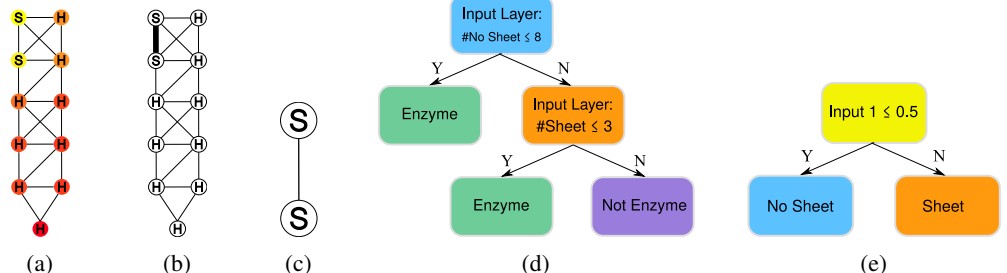

Figure 1: PROTEINS is a graph classification dataset. Nodes are secondary structural elements of amino acids, either helixes (H, input 0), sheets (S, input 1), or turns (T, input 2). Graphs are classified whether they are an enzyme or not. (a) shows a node-level explanation for an Enzyme; (b) an edge-level explanation; (c) a subgraph explanation. With all three of these prior approaches, we can deduce that the two sheets are important, but we do not know why. The explanation does not answer questions such as "Do the sheets have to be connected?" or "Must there be exactly two? (d) The GraphChef recipe for PROTEINS reads as follows: *A protein is an enzyme if it has at most* 8 *nodes that are not sheets, or if it has at most* 3 *sheets.* The example graph has 9 "no sheet" and 2 sheet nodes and is therefore an Enzyme. The second tree (e) explains the terms used in (d). Note that this recipe simultaneously explains the predictions for all the other graphs in the dataset. This recipe using only node features and no message passing matches the current approach in biochemistry (Errica et al., 2020).

- We test GraphChef on established GNN explanation benchmarks and real-world graph datasets. We show that our recipes retain high accuracy compared to fully-expressive GNNs. The importance propagation produces competitive results to existing explanation methods. We further validate that the proposed pruning method considerably reduces tree sizes. Last, we demonstrate how to read GraphChef's recipes to find interesting insights in real-world datasets or flaws in existing explanation benchmarks.

- We provide a user interface for GraphChef.[1] This tool allows for the interactive exploration of the GraphChef recipes on the datasets examined in this paper. We provide a manual for the interface in Appendix F.

## 2 RELATED WORK

### 2.1 EXPLANATION METHODS FOR GNNS

Recent years have seen many GNN explanation methods being proposed using different ideas. We can roughly group them into five groups based on the main approach. **Gradient**, **Mutual-Information**, and **Counterfactual** methods compute node-level or edge level importance that we can interpret as heatmaps. These heatmaps highlight which parts of the input are important, but not why. On the other hand, **Subgraph** and **Example** based methods explain graphs by showing other example graphs or idealized prototype representatives for classes—but again not why these (sub)graphs influence the GNN prediction. In all five cases, a human would need explanations for dozens of example graphs to puzzle together a recipe for a dataset.

**Gradient based.** Baldassarre and Azizpour (2019) and Pope et al. (2019) show that it is possible to adopt gradient-based methods known from computer vision, for example Grad-CAM(Selvaraju et al., 2017). Gradients can be computed on node features and edges (Schlichtkrull et al., 2021).
**Mutual-information based.** Ying et al. (2019) measure the importance of edges and node features. Edges are masked with continuous values. Instead of gradients, the authors use mutual information between the inputs and the prediction to quantify the importance. Luo et al. (2020) follow a similar idea but emphasize finding structures that explain multiple instances at the same time.
**Counterfactual.** Lucic et al. (2021) show that already the deletion of a few edges can change the classifier prediction, supporting that these edges are important for the class. The idea is similar to

---

[1] https://interpretable-gnn.netlify.app/

occluding parts of an image in computer vision (Zeiler and Fergus, 2014). Bajaj et al. (2021) propose a hybrid with an example-based explanation. They compute decision boundaries over multiple instances to find optimized counterfactual explanations.

**Subgraph based.** Yuan et al. (2021) consider each subgraph as a possible explanation. To score a subgraph, they use Shapley values (Shapley, 1953) and Monte Carlo tree search for guiding the search. Duval and Malliaros (2021) build subgraphs by masking the nodes and edges in the graph. They run their subgraph through the trained GNN and try to explain the differences to the entire graph with simple interpretable models and Shapley values. Zhang et al. (2021) infer subgraphs called prototypes that each represent one particular class. Graphs are classified and explained through their similarity to the prototypes. Azzolin et al. (2022) propose a scheme to combine "local" subgraphs explaining one graph into a "global" logic formula to explain the reasoning in the dataset.

**Example based.** Huang et al. (2020) proposes a graph version of the LIME (Ribeiro et al., 2016) algorithm. A prediction is explained through a linear decision boundary built by close-by examples. Vu and Thai (2020) aim to capture the dependencies in node predictions and express them in probabilistic graphical models. Faber et al. (2020) explain a node by giving examples of similar nodes with the same and different labels. Dai and Wang (2021) create a $k$-nearest neighbor model and measure similarity with GNNs. Yuan et al. (2020a) and Wang and Shen (2022) generate a representative graph for each class that maximizes the model's confidence in the class prediction. (Azzolin et al., 2022) is a noteworthy approach that constructs formulas that aim not to explain single graphs but also classes for the entire dataset, being the most similar to our dataset-level recipes. However, these formulas do not reveal the GNN decision process either.

**Simple GNNs.** Another interesting line of research is simplified GNN architectures Cai and Wang (2018); Chen et al. (2019); Huang et al. (2021). The main goal of this research is to show that simple architectures can perform competitively with traditional complex GNNs. As a side effect, the simplicity of these architectures also makes them slightly more understandable. However, they are not understandable to the extent that we can derive recipes for entire datasets. GraphChef also deliberately sacrifices some expressive power but explicitly to gain explainability.

## 2.2 EXPLANATION PROPERTIES AND BENCHMARKS.

Complimentary to the development of explanation methods is research on how to evaluate these methods. Sanchez-Lengeling et al. (2020) and Yuan et al. (2020b) discuss the desirable properties a good explanation method should have. For example, an explanation method should be faithful to the model, which means that an explanation method should reflect the model's performance and behavior. Agarwal et al. (2022) provide a theoretical framework to define how strong explanation methods adhere to these properties. They also derive bounds for several explanation methods. Faber et al. (2021) and Himmelhuber et al. (2021) discuss deficiencies in the existing benchmarks used for empirical evaluation. We will show recipes by GraphChef how GNNs might exploit such deficiencies in Appendix A to produce correct recipes that are not in line with the explanation ground truth.

## 2.3 COMBINING DECISION TREES WITH NEURAL NETWORKS

Decision trees are popular machine learning models thanks to their inherent explainability (given reasonable tree sizes). Already early works in neural networks research investigated the feasibility of extracting trained neural networks into decision trees to understand what the network learned (Boz, 2002; Craven and Shavlik, 1995; Dancey et al., 2004; Krishnan et al., 1999).

Recently, this idea has been picked up again. Schaaf et al. (2019) have shown that encouraging sparsity and orthogonality in neural network weight matrices allows for model distillation into smaller trees with higher final accuracy. Wu et al. (2017a) follow a similar idea for time series data: they regularize the training process for recurrent neural networks to penalize weights that cannot be easily modeled by decision trees. Yang et al. (2018a) aim to directly learn neural trees. Their neural layers learn how to split the data and put it into bins. Stacking these layers creates trees. Kontschieder et al. (2015) learn neural decision forests by making the routing in nodes probabilistic and learning these probabilities and leaf predictions. A recent work by Aytekin (2022) shows that we can transform any neural network into decision trees. However, this approach creates a tree with potentially exponentially many leaves. Even though this method produces decision trees, we cannot use the outputs as humanly understandable recipes for datasets.

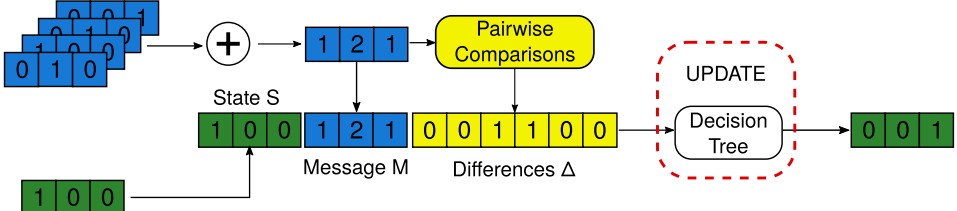

Figure 2: A GraphChef layer. GraphChef updates the state of a node based on its previous categorical state (left green), the number of neighbors per state (blue, 1 in state 0, 2 in state 1, 1 in state 2), and binary $>$ comparisons between states (yellow, only state 1 outnumbers other states, therefore the third and fourth deltas are 1). A decision tree that receives this information computes the followup categorical state (right green).

GraphChef follows the same underlying idea. Can we express what our GNN learned in a decision tree to explain its decision process? In contrast to existing methods which operate on tabular data, GraphChef also needs to include the reasoning about the graph structure (for example, finding important edges between nodes). We also emphasize the importance of small trees to ensure that the recipes produced are understandable to humans.

## 3 THE GRAPHCHEF MODEL

### 3.1 FROM GIN TO DISH

We follow the general message passing GNN model (Gilmer et al., 2017; Battaglia et al., 2018): Every node has an internal state that is modified in the GNN layers. In every layer, every node computes a message that it sends to every neighbor. Then every node receives all messages, aggregates the set of messages, and updates its state:

$$h_v^{l+1} = \text{UPDATE}_\theta(h_v^l, \text{AGGREGATE}\left\{\text{MESSAGE}(h_w^l)\right\}_{w \in Nb(v)}).$$

Our starting architecture is the GIN model from Xu et al. (2019). In GIN, the MESSAGE function is the identity function, AGGREGATE is element-wise summation, and UPDATE is a learnable neural function $f_\theta$. Although it is rather simple, the GIN model is already as expressive as the $1-$Weisfeiler Lehman test. We can notate GIN as follows:

$$h_v^{l+1} = f_\theta^l(h_v^l, \sum_{w \in Nb(v)} h_w^l).$$

We derive the internal state $h_v^0$ for the first dish layer with an encoder on the initial node features $x_v$. We use the internal states in a decoder layer: For node classification, we use skip connections and use all internal states for the final prediction. For graph classification, we sum-pool all nodes in every layer and make sums available for the final prediction.

The internal states are $d-$dimensional real-valued vectors: $h_v^l \in \mathbb{R}^d$. These vectors allow complex relationships between features that make explainability very difficult. We simplify the internal states by applying a Gumbel-Softmax (Jang et al., 2016; Maddison et al., 2016) to the GNN layers and also to the encoder layer:

$$h_v^{l+1} = Gumbel(f_\theta^l(h_v^l, \sum_{w \in Nb(v)} h_w^l)).$$

Therefore, the internal states $h_v^l$ become one-hot categorical values and summation in the GNN aggregation step counts how many neighbors are in what state. The theoretical motivation for this change comes from distributed computing: Loukas (2020) showed that message passing GNNs such as GIN are equivalent to the LOCAL distributed computing model(Peleg, 2000). In LOCAL, nodes can perform arbitrary local computation. However, there exists also a simpler model, coined the stone-age model (Emek and Wattenhofer, 2013) where nodes can only transition between categorical states using a finite state machine. The nodes can count the number of neighbors in each state and only

in a limited manner in the spirit of "one, two, three, many". Neighborhood counts above a threshold are indistinguishable from each other. Interestingly enough, such a simplified model can still solve many distributed computing problems. Our proposed GNN architecture is the GNN equivalent of this stone-age model. Therefore, we coin our layer "dish": differentiable stone-"h".

Let us look at the theoretical expressive power of dish layers: For a GIN with $\log(d)$ bits of continuous embedding space, we can, in principle, create a dish GNN with $d$ categorical states and the same theoretical expressiveness. Practically, we aim for a low number of categorical bits to ensure human interpretability. We investigate the drop in accuracy in Table 1a.

### 3.2   FROM DISH TO GRAPHCHEF

We can leverage the categorical states of a trained dish model to distill all neural blocks to decision trees. These are the update functions in the dish layers as well as the functions in the encoder and decoder layers. Since all states are categorical, this distillation becomes a classification problem where decision trees learn to predict the categorical state. A distilled GraphChef layer looks as follows and is also shown in Figure 2:

$$h_v^{l+1} = TREE^l(h_v^l, \sum_{w \in Nb(v)} h_w^l).$$

Note that the GraphChef layer still follows the message passing framework but instead of a neural function plus a Gumbel-Softmax it uses a decision tree. We empirically found that GraphChef benefits from one tweak. Decision trees generally struggle with comparing two features (is one feature larger than the other). To help with that and produce small trees, we include pairwise delta features $\Delta$. These binary features compare every ordered pair of features and are one if the first feature is larger. Let $c_i^l$ be the counts of neighbors in state $i$ in layer $l$:

$$\Delta(c_v^l) = \underset{i \in S, j \neq i \in S}{\|} \mathbb{1}_{c_i^l > c_j^l}$$
$$h^{l+1}(v) = TREE^l(h_v^l, c_v^l, \Delta(c_v^l)).$$

Decision trees have access to three sets of features: the previous state, the number of neighbors in each state or the comparison of two state counts. Figure 2 shows these colored green, blue, and yellow, respectively. For each decision node in GraphChef's decision trees, we can interpret the GNN reasoning based on which feature is used as shown in 3. We can combine these individual node interpretations to formulate GraphChef's recipes. Figure 5 shows an example recipe with an interpretation in Table 4. More examples of recipes are available in Appendix A.

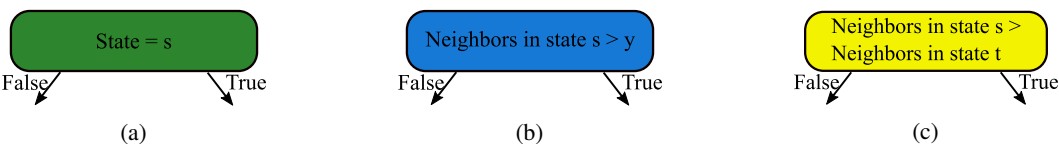

(a)                                                 (b)                                                 (c)

Figure 3: The different branches possible in a GraphChef layer. We can branch on (a) the state in which a node is in, (b) if the node has a certain number of neighbors in a certain state, or (c) if the node has more neighbors in one state than in another state. The colors match the features in Figure 2.

### 3.3   PRUNING GRAPHCHEF

Although sufficiently deep decision trees can be universal function approximators(Royden and Fitzpatrick, 1988; Aytekin, 2022), we prefer small and shallow decision trees, which are much more understandable to humans. Shallow trees are more akin to the finite state machine used in the stone-age distributed computing model. We find that setting an upper bound on leaves for every decision tree is not sufficient and that GraphChef requires further pruning.

We prune these nodes based on the reduced error pruning algorithm (Quinlan, 1987). First, we define a pruning set. The validation set alone is too small to cover all paths in the trees, which causes over-pruning. On the other hand, we cannot use only the training set since this set created the overfitting artifacts. We propose merging both sets for our pruning set.

We also need a quality criterion when replacing an inner decision node with a leaf. We propose the following: If replacing a node with a leaf (i) does not drop the accuracy on the validation set and (ii) does not drop the training accuracy below the validation accuracy, we accept the replacement. Not allowing validation accuracy to drop ensures that we do not over-prune. However, allowing drops in training accuracy allows for removing decision nodes that result from overfitting. Not allowing training accuracy to drop below validation accuracy is another safeguard against over-pruning. We keep iterating over all inner decision nodes, sorted by the number of data points they cover, and try replacing them with leaf nodes until we find no more nodes that we can drop without accuracy loss. Afterwards we keep iterating and remove the node with the smallest drop in validation accuracy. Slight deterioration allows to prune far more nodes. Ultimately, we record 10 pruning levels: after pruning with no validation accuracy loss and pruning the remaining nodes in $10\%$ steps. We make these levels available in the UI and allow users to choose the trade-off between accuracy and tree size.

**Computing explanation scores.** If we have a GraphChef recipe for the dataset, we can also compute heatmap-style importance scores for single graphs; similar to existing graph explanation methods. We compute these importance scores layer by layer. In the input layer, every node is its own explanation. In each GraphChef layer, we compute Tree-Shap values (Lundberg et al., 2018) for every decision tree feature. We then compute importance updates for every decision tree feature weighted by this Tree-Shap value independently (unused features have a value of $0$). If the node uses a state feature (as in Figure 3a) then we give importance to the node itself. If we use a message feature (as in Figure 3b), we distribute the importance evenly between all neighbors in this state. If we use a delta feature (as in Figure 3c), we distribute positive importance between all neighbors in the majority class and also negative importance between all neighbors in the minority class. Finally, we normalize the scores to sum up to $1$. In the decoder layer, we employ skip connections to also consider intermediate states (for node classification) or intermediate pooled node counts (for graph classification). We provide a formal computation in Appendix C.

## 4 EXPERIMENTS

|  |  |  |  |  | GraphChef | |
| --- | --- | --- | --- | --- | --- | --- |
| Dataset | DT | DT+degrees | GIN | dish GNN | No pruning | Lossless pruning |
| Infection | 0.43±0.00 | 0.43±0.00 | 0.98±0.04 | 1.00±0.00 | 1.00±0.00 | 1.00±0.00 |
| Negative | 0.51±0.00 | 0.50±0.00 | 1.00±0.00 | 1.00±0.00 | 1.00±0.00 | 1.00±0.00 |
| BA-Shapes | 0.43±0.00 | 0.86±0.02 | 0.97±0.02 | 1.00±0.01 | 0.99±0.01 | 0.99±0.01 |
| Tree-Cycles | 0.59±0.00 | 0.84±0.04 | 1.00±0.00 | 1.00±0.00 | 1.00±0.00 | 1.00±0.00 |
| Tree-Grid | 0.58±0.00 | 0.76±0.03 | 1.00±0.01 | 0.99±0.01 | 0.99±0.01 | 0.99±0.01 |
| BA-2Motifs | 0.50±0.00 | 0.82±0.03 | 1.00±0.00 | 1.00±0.00 | 1.00±0.00 | 1.00±0.00 |
| MUTAG | 0.83±0.09 | 0.85±0.07 | 0.88±0.05 | 0.88±0.06 | 0.88±0.06 | 0.85±0.08 |
| Mutagenicity | 0.71±0.02 | 0.72±0.02 | 0.81±0.02 | 0.79±0.02 | 0.75±0.02 | 0.74±0.02 |
| BBBP | 0.83±0.03 | 0.83±0.02 | 0.81±0.04 | 0.83±0.03 | 0.82±0.03 | 0.83±0.03 |
| PROTEINS | 0.74±0.02 | 0.71±0.04 | 0.70±0.03 | 0.71±0.02 | 0.71±0.04 | 0.71±0.04 |
| IMDB-B | 0.57±0.04 | 0.71±0.04 | 0.69±0.04 | 0.70±0.05 | 0.69±0.03 | 0.69±0.04 |
| REDDIT-B | 0.75±0.03 | 0.80±0.03 | 0.87±0.10 | 0.90±0.03 | 0.88±0.03 | 0.87±0.04 |
| COLLAB | 0.59±0.01 | 0.70±0.02 | 0.72±0.01 | 0.70±0.02 | 0.69±0.02 | 0.69±0.02 |

(a)

Table 1: a) Test set accuracies using decision trees (DT) and different GNN layers. One version of decision trees additionally receives the degrees. For graph classification, the decision tree inputs where the number of nodes of each type and potentially of each degree. The gap between dishGNN and both GraphChef versions to a full GNN is low, importantly also in graphs that require sophisticated graph reasoning (and DT+degrees is not enough)

### 4.1 EXPERIMENT SETUP

**Datasets.** We first run GraphChef on synthetic GNN explanation benchmarks introduced in previous work. We use the Infection and Negative Evidence benchmarks from Faber et al. (2021), The BA-Shapes, Tree-Cycle, and Tree-Grid benchmarks from Ying et al. (2019), and the BA-2Motifs dataset from Luo et al. (2020). Second, we experiment with the following real-world datasets: MUTAG (Debnath et al., 1991); BBBP (Wu et al., 2017b); Mutagenicity (Kazius et al., 2005); PROTEINS, REDDIT-BINARY, IMDB-BINARY, and COLLAB (Borgwardt et al., 2005). We provide

more information for all datasets, such as statistics, descriptions, and examples in Appendix D and hyperparameters in Appendix E. Note that all datasets except COLLAB are small enough to train on commodity CPUs. For example, training the PROTEINS dataset for one seed on a laptop trains in 5 minutes for the full 1500 epochs, a few seconds for the tree distillation, and $1 - 2$ minutes for tree pruning. The larger REDDIT-BINARY takes around one hour to train a dish GNN (if it uses all epochs) a few seconds for distilling trees and around 10 minutes for pruning. Computing lossy pruning takes a comparable amount of time to lossless pruning.

## 4.2 QUANTITATIVE RESULTS

| Method | Infection | Saturation | BA-Shapes | Tree-Cycles | Tree-Grid |
|---|---|---|---|---|---|
| Gradient | 1.00±0.00 | 1.00±0.00 | 0.882 | 0.905 | 0.667 |
| GNNExplainer | 0.32±0.09 | 0.32±0.05 | 0.925 | 0.948 | 0.875 |
| PGMExplainer | 0.38±0.06 | 0.01±0.01 | 0.965 | 0.968 | 0.892 |
| GraphChef | 0.95±0.02 | 1.00±0.00 | 0.94±0.02 | 0.84±0.02 | 0.927±0.01 |

Table 2: Overlap of identified explanation to explanation ground truth. The numbers for Gradient, GNNExplainer, and PGMExplainer are taken from Ying et al. (2019), Vu and Thai (2020), and Faber et al. (2021).

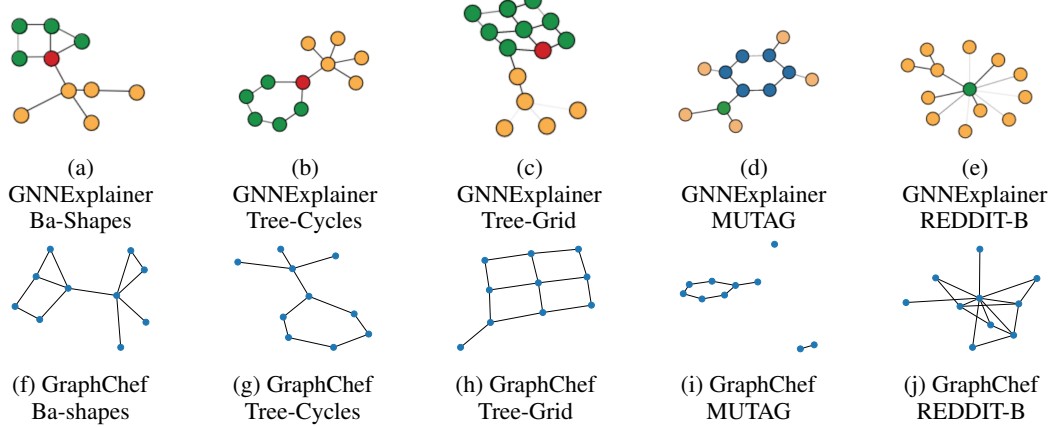

| (a) GNNExplainer Ba-Shapes | (b) GNNExplainer Tree-Cycles | (c) GNNExplainer Tree-Grid | (d) GNNExplainer MUTAG | (e) GNNExplainer REDDIT-B |
|---|---|---|---|---|
| (f) GraphChef Ba-shapes | (g) GraphChef Tree-Cycles | (h) GraphChef Tree-Grid | (i) GraphChef MUTAG | (j) GraphChef REDDIT-B |

Figure 4: Subselections of important nodes for GNNExplainer and GraphChef for several datasets.

**GraphChef performs comparably to GIN.** On both groups of datasets, we measure the performance of GIN versus dish GNN versus GraphChef. In principle, there may be a drop in accuracy at each step to trade expressive power for explainability. Table 1a shows the average test accuracy over 10 seeds for each method and also for the losslessly pruned version of GraphChef. We find that GraphChef recipes perform very close to GIN. This also holds on datasets that require sophisticated graph reasoning that we cannot solve by a decision tree baseline that has access to degrees. The model simplifications to obtain understandable recipes do not decrease accuracy. We observe that tree pruning can even have a *positive* effect on test accuracy compared to non-pruned GraphChef. This is likely due to the regularization induced by the pruning procedure.

**GraphChef produces competitive explanations.** We can use the ground truth available from the first group of datasets to evaluate the importance scores of GraphChef (Appendix C). Following existing work such as Ying et al. (2019), we compute importance scores for every node in the graph and consider the $n$ nodes with the highest score for the explanation, $n$ is the number of nodes in the ground truth. The explanation accuracy for a graph is the ratio of correct nodes in the explanation. Table 2 shows the average accuracy per dataset.

GraphChef explanations are competitive with existing explanation methods with some examples shown in Figure 4. The scores on Tree-Cycles and Tree-Grid suggest room for improvement, but

that is not actually the case as we show in Appendix A. GraphChef solves the datasets in a legitimate way that is easier than the creators anticipated and does not require finding the full motif. These alternative solutions are known deficiencies of the datasets (Faber et al., 2021; Himmelhuber et al., 2021).

| Dataset | No pruning | | REP Training | | REP Validation | | REP Ours | | REP Lossy | |
|---|---|---|---|---|---|---|---|---|---|---|
| | Accuracy | Size | Accuracy | Size | Accuracy | Size | Accuracy | Size | Accuracy | Size |
| Infection | 1.00±0.00 | 205±56 | 1.00±0.00 | 26±2 | 1.00±0.00 | 25±2 | 1.00±0.00 | 26±2 | 0.98±0.01 | 17±2 |
| Negative | 1.00±0.00 | 18±14 | 1.00±0.00 | 5±0 | 1.00±0.00 | 5±0 | 1.00±0.00 | 5±0 | 1.00±0.00 | 4±0 |
| BA-Shapes | 0.99±0.01 | 30±10 | 0.99±0.01 | 21±5 | 0.97±0.03 | 15±4 | 0.99±0.01 | 21±5 | 0.98±0.04 | 17±4 |
| Tree-Cycles | 1.00±0.00 | 19±5 | 1.00±0.00 | 11±3 | 0.99±0.02 | 9±2 | 1.00±0.00 | 11±3 | 0.99±0.01 | 9±3 |
| Tree-Grid | 0.99±0.01 | 30±13 | 0.99±0.01 | 17±8 | 0.99±0.01 | 13±4 | 0.99±0.01 | 15±8 | 0.99±0.01 | 15±8 |
| BA-2Motifs | 1.00±0.00 | 141±43 | 1.00±0.00 | 12±3 | 1.00±0.01 | 11±3 | 1.00±0.00 | 13±4 | 1.00±0.00 | 13±4 |
| MUTAG | 0.88±0.06 | 59±27 | 0.86±0.08 | 19±17 | 0.83±0.07 | 7±6 | 0.85±0.08 | 18±16 | 0.85±0.08 | 18±16 |
| Mutagenicity | 0.75±0.02 | 375±13 | 0.76±0.02 | 154±19 | 0.73±0.01 | 56±16 | 0.74±0.02 | 91±36 | 0.73±0.02 | 50±19 |
| BBBP | 0.82±0.03 | 366±53 | 0.84±0.02 | 88±52 | 0.79±0.04 | 8±10 | 0.83±0.03 | 46±27 | 0.82±0.03 | 31±18 |
| PROTEINS | 0.71±0.04 | 206±90 | 0.72±0.03 | 12±13 | 0.70±0.04 | 8±6 | 0.71±0.04 | 9±6 | 0.71±0.04 | 9±6 |
| IMDB-B | 0.69±0.03 | 218±32 | 0.69±0.04 | 20±9 | 0.66±0.06 | 16±6 | 0.69±0.04 | 29±9 | 0.69±0.04 | 29±9 |
| REDDIT-B | 0.88±0.03 | 248±28 | 0.88±0.02 | 53±14 | 0.85±0.04 | 28±8 | 0.87±0.04 | 49±21 | 0.87±0.04 | 38±15 |
| COLLAB | 0.69±0.02 | 301±1 | 0.70±0.02 | 36±15 | 0.67±0.03 | 22±12 | 0.69±0.02 | 30±18 | 0.68±0.02 | 21±12 |

Table 3: Running reduced error pruning (REP) on different pruning sets. The threshold for lossy pruning is chosen manually, scores for the remaning considered thresholds are visible in the UI.

**Pruning significantly reduces the decision tree sizes.** Third, we examine the effectiveness of our pruning method. We compare the tree sizes before pruning, after lossless pruning, and after lossy pruning. We measure tree size as the sum of decision nodes over all trees. Additionally, we verify the effectiveness of using our pruning criterion for reduced error pruning and compare it against simpler setups of using only the training or validation set for pruning. We report tree sizes and test set accuracy for all configurations in Table 3.

We can see that reduced error pruning leads to an impressive drop in the number of nodes required in the decision trees. On average, we can prune about 62% of nodes in synthetic datasets and even around 84% of nodes in real-world datasets without a loss in accuracy. If we accept small drops in accuracy, we can even prune a total of 68% and 87% of nodes in synthetic and real-world datasets, respectively. Among the different setups for reduced error pruning, our proposed approach of using both training and validation accuracy performs the best. As expected, pruning only on the validation set tends to over-prune the trees: Trees become even smaller, but there is also a larger drop in accuracy, especially in the real-world datasets. Using the training set leads to underpruning; there is no drop in accuracy, but decision trees for real-world graphs tend to stay large. Lossy pruning usually allows further decrease in decision tree sizes with virtually no accuracy drop. In this instance, we used the UI (Appendix F) to choose a threshold. Alternative options are also available in the UI.

## 4.3 QUALITATIVE RESULTS

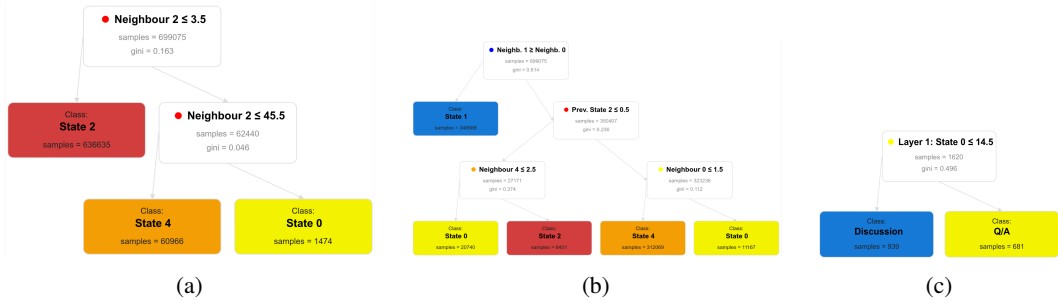

(a) (b) (c)

Figure 5: GraphChef recipe for the Reddit-Binary dataset. Decision trees in every layer are built using the building blocks from Figure 3. Table 4 provides an interpretation of all states in all layers and for the entire dataset.

| Layer | State | Decision Rule | Interpretation |
|---|---|---|---|
| Encoder | 2 | All nodes | No differentiation due to no features. |
| Layer 0 | 2 | Nodes with at most 3 neighbors | Inactive users |
| Layer 0 | 4 | Between 4 and 45 neighbors | Active users |
| Layer 0 | 0 | More than 45 neighbors | Central users |
| Layer 1 | 0 | 1) State 0 nodes with more than one state 0 neighbor or 2) Not state 2 nodes that have at most two state 4 neighbors | 1) Inactive users writing with at least 2 central users or 2) Active or central users that write with at most 2 active users. |
| Layer 1 | 1 | No neighbor in state 0 | Users that do no write with a central user. |
| Layer 1 | 2 | Not state 2 nodes with at least 3 state 4 neighbors | Active or central users that write with at least 3 active users. |
| Layer 1 | 4 | Nodes in state 2 with exactly one state 0 neighbor | Inactive users write with one central user. |
| Decoder | Q/A | At least 15 nodes in Layer 1 state 0 | See interpretation of Layer 1 state 0. |
| Decoder | Discussion | Otherwise | The GraphChef model looks for evidence of a Q/A graph. Discussions are "not Q/A" graphs. |

Table 4: Analysis of GraphChef recipe in Figure 5 for the Reddit-Binary dataset. A Q/A graph requires central users and 15 users that 1) are inactive and write with more than one central user or 2) are active or central and write mostly with inactive users.

Finally, let us look at how to read a GraphChef recipe for the Reddit-Binary dataset. The recipe is shown in Figure 5. First, we aim to understand every categorical state in every layer, in a fashion similar to dynamic programming. We start by understanding the states in the first layer, taking note of these explanations, and using them to understand the next layer. Table 4 does this for the Reddit-Binary dataset from top to bottom. To understand the dataset, we inspect the decoder rules.

We can understand that we need to find a certain amount of users (15) fulfilling certain conditions: 1) inactive users writing with at least two central users or 2) active or central users that write with at least one central user and at most 2 active users. We can understand 1) as just replying to a single central user is not sufficient. We hypothesize that controversial opinions in discussions can also attract many comments, even by inactive users. Users fulfilling 2) write mostly with inactive users since there are few central users and little communication with active users is allowed.

GraphChef's recipe aligns with our belief that Q/A graphs are more "star-like" than discussion graphs (Faber et al., 2020). However, the recipe also defines what we should consider as "star like". We obtain a threshold on what we can consider as centers of the star (central nodes with a degree of 46 or higher). The recipe also tells us to what extent non-star communication is acceptable for a Q/A graph (two active users). To the best of our knowledge, such insight about the Reddit-Binary dataset has not been found yet with existing explanation methods. See Appendix A for more recipe analyses.

## 5    CONCLUSION, LIMITATIONS, AND FUTURE WORK

In this paper, we introduce GraphChef, a new architecture that takes GNN explanations to a new level. Instead of only highlighting the important parts of an input, GraphChef produces a recipe which reveals the full decision process for the whole dataset. Internally, GraphChef combines GNNs and decision trees to create recipes. We believe that GraphChef will help improve our understanding of graph problems, which is crucial for the adoption of GNNs in safety critical domains such as medicine. We can also identify and reject recipes that make biased or discriminatory decisions.

As a limitation, we found that GraphChef can struggle to create recipes for datasets with a large feature space, discussing details in Appendix B. It seems difficult to construct small and accurate decision trees when the input feature space is very large (hundreds or thousands of features).

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

# A    MORE GRAPHCHEF RECIPE ANALYSES

## A.1    MUTAG

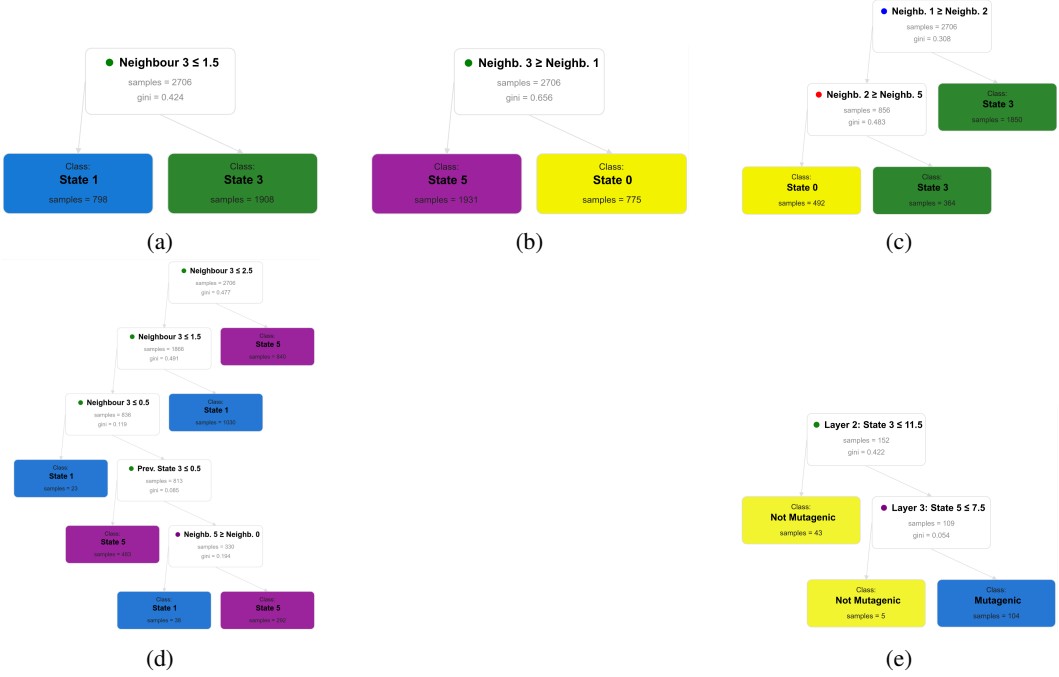

Figure 6: GraphChef recipe for MUTAG. Table 5 shows an interpretation for all states in all layers. For a graph to be mutagenic, it requires at least twelve atoms other than O and eight atoms that 1) have three or more non-O bindings 2) are O atoms, 3) bound to $O_2$ atoms.

Table 5 shows an incremental interpretation (from Encoder to Decoder) of the states in all layers that are shown in the trees the GraphChef recipe in Figure 6 for the MUTAG dataset. The encoder shows that graph size is important (we need at least twelve non-O atoms) and O atoms and their connectivity play a role. We need at least eight nodes that 1) have three non-O bindings 2) are O atoms 3) are connected to $O_2$ groups. The last two conditions highlight why $NO_2$ groups are associated with mutagenicty: The N atom fulfills the last condition and the two O atoms fulfill the other condition. However, in the MUTAG (Debnath et al., 1991) dataset, these structures are not sufficient for mutagenic molecules.

| Layer | State | Decision Rule | Interpretation |
|---|---|---|---|
| Encoder | 3 | All nodes receive state 3 | GraphChef drops atom types (and rediscovers them later via degrees). |
| Layer 0 | 1 | Less than two neighbors | Degree 1 nodes, H atoms are implicit so these represent O atoms. |
| Layer 0 | 3 | At least two neighbors | Nodes with at least two electron bindings, predominantly C and N. |
| Layer 1 | 0 | More neighbors in state 1 than state 3 | Atoms with majorly bindings to O atoms. |
| Layer 1 | 5 | At least as many state 3 as state 1 neighbors | Atoms not connected to O, or mainly to other atom types. |
| Layer 2 | 0 | No neighbor in state 5 | Rediscovers almost all O atoms, especially those in $O_2$ groups. They have one neighbor each and that neighbor has bindings to mostly O. |
| Layer 2 | 3 | At least one neighbor in state 5 | Atoms other than O and some O atmons that are not in $O_2$ groups. |
| Layer 3 | 1 | 1) Exactly two state 3 neighbors 2) Nodes with no state 3 neighbors 3) Nodes with 1 state 3 neighbor but at most state 0 neighbor | 1+3) Nodes with at most one O neighbor and at most two other neighbors 2) Nodes with only O neighbors. |
| Layer 3 | 5 | 1) Nodes with at least 3 state 3 neighbors 2) not state 3 nodes 3) state 3 nodes with one state 3 and 2 state 0 neighbors | 1) Atoms with at least 3 connections to atoms other than O 2) O atmons 3) atoms connected to $O_2$ groups. |
| Decoder | Mutagenic | At least twelve atoms in layer 2 state 3 and at least eight nodesin layer 3 state 5. | At least twelve atoms other than O and 1) Atoms with at least 3 connections to atoms other than O 2) O atmons 3) atoms connected to $O_2$ groups. |
| Decoder | Not Mutagenic | otherwise | otherwise |

Table 5: Analysis of the GraphChef recipe in Figure 6 for the MUTAG dataset. For a graph to be mutagenic, it requires at least twelve atoms other than O and eight atoms that 1) have three or more non-O bindings 2) are O atoms, 3) bound to $O_2$ atoms.

## A.2 BA-2MOTIFS

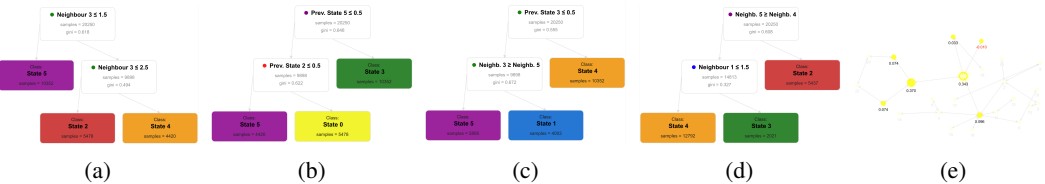

| | (a) | (b) | (c) | (d) | (e) |

Figure 7: Layers of GraphChef for the decision process on BA-2MOTIFS. Table 6 shows an interpretation for all states in all layers. The model learns to identify house nodes and classify such graphs. Cycle graphs are graphs which are not house graphs, thus solved with the bias term. Explanation scores for cycles are therefore off (e).

| Layer | State | Decision Rule | Interpretation |
|---|---|---|---|
| Encoder | 3 | All nodes | No node features available for differentiation. |
| Layer 0 | 2 | Two state 3 neighbors | Degree 2 nodes. |
| Layer 0 | 4 | Three or more state 3 neighbors | Degree 3 or higher nodes. |
| Layer 0 | 5 | Less than two neighbors | Degree 1 nodes (graphs are connected). |
| Layer 1 | 0 | State 2 nodes | Degree 2 nodes. |
| Layer 1 | 3 | State 5 nodes | Degree 1 nodes. |
| Layer 1 | 5 | Neither state 2 nor 5 | Degree 3 or higher nodes. |
| Layer 2 | 1 | Not state 3 nodes with more state 5 than state 3 neighbors. | House candidates: nodes with a degree of at least 2, with at least one degree 3 or higher neighbor and no degree 1 neighbor. |
| Layer 2 | 4 | State 3 nodes | Degree 1 nodes. |
| Layer 2 | 5 | Not state 3 nodes with state 5 neighbors | At least degree 2 nodes but connected to at least degree 1 neighbor (which house nodes do not have). |
| Layer 3 | 2 | More state 4 than state 5 neighbors | Nodes that have majorly degree 1 neighbors (not house nodes). |
| Layer 3 | 3 | At least two state 1 neighbors | House nodes: connected to two more house candidates. |
| Layer 3 | 4 | At most one state 1 neighbor | Nodes connected to at most one house candidate (wrong for every node in the house). |
| Decoder | House | At least five nodes in layer 3 state 3. | Graphs with at least five house nodes. |
| Decoder | Cycle | otherwise | otherwise. |

Table 6: Analysis of the GraphChef recipe in Figure 7 for the BA-Motifs dataset. The model learns to identify house nodes and classify such graphs. Cycle graphs are graphs which are not house graphs, thus solved with the bias term.

Table 6 shows an interpretations of the states in all layers for the GraphChef recipe for BA-2MOTIFS. Figure 6 shows the recipe. GraphChef only learns to identify house nodes. The important step is state 1 in the second layer. Due to the Barabasi-Alert base graph structure, house nodes stand out with their degree of 2 or 3. The next layer confirms the house as house candidates that are connected to house candidates. The model does not learn about cycles at all, Cycles are "not houses". We can see that GraphChef found and exploited the pitfall about bias terms noted by Faber et al. (2020); Himmelhuber et al. (2021). Figure 7e shows that we cannot trust explanation scores for cycles.

## A.3 TREE CYCLE

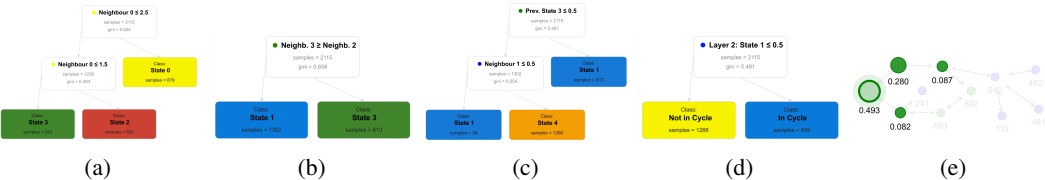

$$\begin{array}{ccccc} (a) & (b) & (c) & (d) & (e) \end{array}$$

Figure 8: GraphChef recipe for TREE-CYCLE. Table 6 shows an interpretation for all states in all layers. A degree check for degree 2 nodes finds the cycles quickly without considering the whole structure. Therefore, the explanation scores for the cycles are off (e).

| Layer | State | Decision Rule | Interpretation |
|---|---|---|---|
| Encoder | 0 | All nodes | No node features available for differentiation. |
| Layer 0 | 0 | Three or more state 0 neighbors | Degree 3 or higher nodes (inner nodes in the tree, cycle node connecting the cycle to the tree). |
| Layer 0 | 2 | Two neighbors in state 0 | Degree 2 nodes (root node and cycles nodes). |
| Layer 0 | 3 | One or zero neighbors in state 0 | Degree 1 nodes (leaves in the connected graph). |
| Layer 1 | 1 | At least as many state 3 as state 2 neighbors | As least as many leaves as cycle neighbors (true for inner nodes as well having zero of both). |
| Layer 1 | 3 | More state 2 than state 3 neighbors | Most cycle nodes, nodes connected to degree 2 nodes. |
| Layer 2 | 4 | Not previous state 3 and at least one state 1 neighbor | Not already a cycle node and connected to a non-cycle node. |
| Layer 2 | 1 | 1) Previous state 3 or 2) no state 1 neighbors | 1) already a cycle node or 2) only connected to cycle nodes. |
| Decoder | Cycle | In layer 2 state 1 | See previous state. |
| Decoder | No Cycle | otherwise | otherwise. |

Table 7: Analysis of the GraphChef recipe in Figure 8 for the Tree-Cycle dataset. The base graph contains only one degree two node that is not part of a cycle. A degree check quickly finds the cycles.

Table 7 shows an interpretations of the states in all layers for the GraphChef recipe for BA-2Motifs. Figure 8 shows the recipe. Due to the base graph being a binary tree, degree two nodes (especially those connected to degree two nodes) are a strong indicator for cycles. For almost all nodes, GraphChef can identify if they are part of the cycle after two layers. Therefore, we do not even need the whole cycle. This is consistent with the previous analysis on Tree-Cycle that the whole cycle is not necessary (Faber et al., 2020; Himmelhuber et al., 2021). Figure 8e shows that we cannot trust explanation scores for cycles since they find only a subset of the motif.

## A.4 TREE GRID

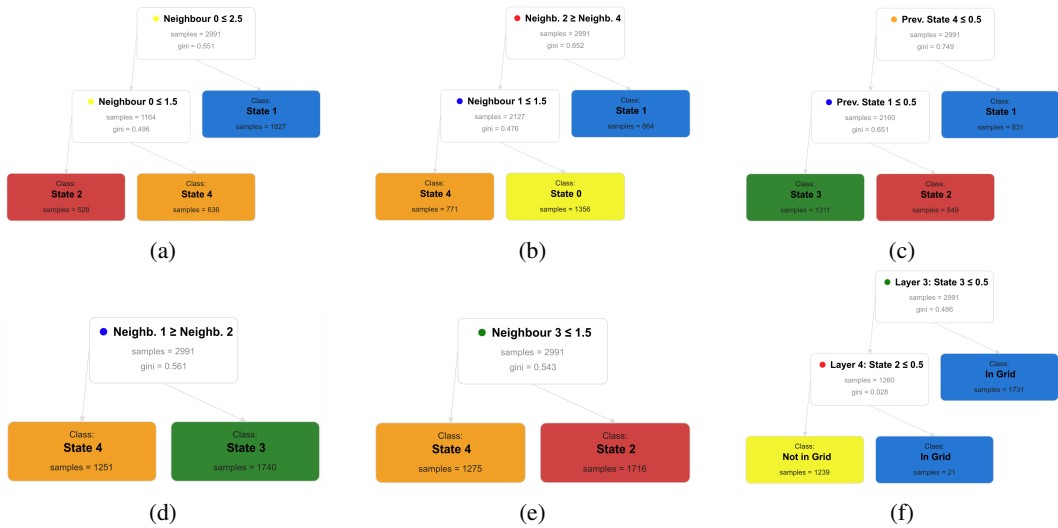

Figure 9: GraphChef recipe for TREE-GRID. Table 8 shows an interpretation for all states in all layers. A degree check for degree 2 nodes find the corner nodes in the grid, after which we explore the remaining motif.

The Tree-Grid (Ying et al., 2019) dataset is similar to the Tree-Cycles dataset we discussed in the main body of the paper. The base graph is a balanced binary tree to which we append $3 \times 3$ grids. As in the Tree-Cycles example, there are (apart from the root node) no other nodes with degree 2 which makes bootstrapping the grid discovery easier. As in the Tree-Cycles example, a GNN does not need to see the whole grid to make a prediction. Table 8 shows the interpretation of the layers and states of GraphChef shown in Figure A.4. The corner nodes in the grid can be quickly found with a degree check. The remaining grid can be found by exploring the neighborhood.

Some nodes can identify that they are part of the grid in just three layers — importantly the corner nodes generally belong to this group. This means that these nodes do not need to consider the opposite corner node at distance $4$. GraphChef does not include this node in its explanation. This is consistent with the explanation accuracy in Table 2: GraphChef achieves a bit more than $\frac{8}{9}$ which means that one node is missing.

| Layer | State | Decision Rule | Interpretation |
|---|---|---|---|
| Encoder | 0 | All nodes | No node features available for differentiation. |
| Layer 0 | 1 | Three or more state 0 neighbors | Degree 3 or higher nodes. (Inner tree nodes and grid nodes except corners) |
| Layer 0 | 2 | Less than two state 0 neighbors | Degree 1 nodes (leaves in the tree). |
| Layer 0 | 4 | Two state 0 neighbors | Degree 2 nodes (root node, corner nodes in the grid). |
| Layer 1 | 0 | At least as many state 2 as state 4 neighbors and at least two state 1 neighbors | Inner nodes in the tree and grid, except parents of leaf nodes. |
| Layer 1 | 1 | More state 4 than state 2 neighbors | Nodes connected to the root or grid corners. |
| Layer 1 | 4 | At least as many state 2 as state 4 neighbors and at most one state 1 neighbors | Leaves and their parent nodes. |
| Layer 2 | 1 | Previous state 4 nodes | Leaves and their parents. |
| Layer 2 | 2 | Previous state 1 nodes | Nodes connected to the root or grid corners. |
| Layer 2 | 3 | Otherwise | Inner nodes in the tree and grid, except parents of leaf nodes. |
| Layer 3 | 3 | More state 2 than state 1 neighbors | More corner and root nodes in the distance 2 (can be the node itself) than leaves. This captures corner and center nodes in the grid |
| Layer 3 | 4 | At least as many state 1 as state 2 neighbors. | Most nodes since they are connected to leaves or not to the root. |
| Layer 4 | 2 | At least two state 3 neighbors | Nodes with two grid neighbors, finding the remaining grid nodes. |
| Layer 4 | 4 | At most one state 3 neighbor | Nodes with at most one grid neighbor. |
| Decoder | Grid | Layer 3 state 3 or layer 4 state 2 | Corner or center grid node or nodes connected to two such nodes. |

Table 8: Analysis of the GraphChef recipe in Figure A.4 for the Tree-Grid dataset. The base graph contains only one degree two node that is not part of a cycle. A degree check for two quickly finds the corner nodes of grids as a starting point to discover the motif.

A.5 BA-SHAPES

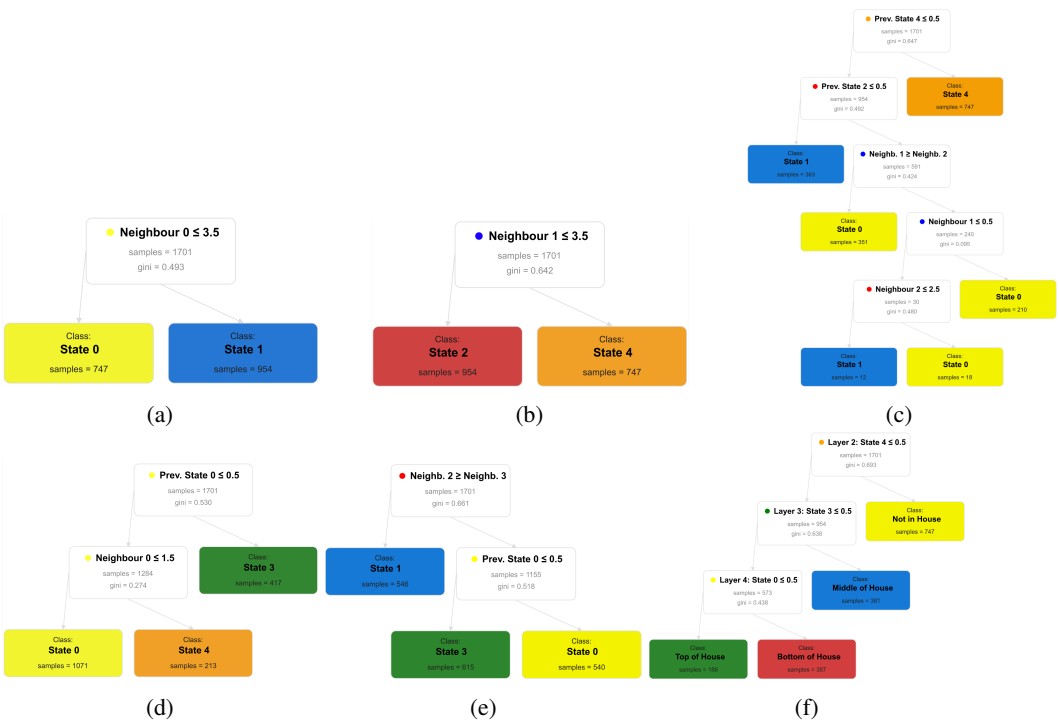

Figure 10: GraphChef recipe for BA-SHAPES. Table 8 shows an interpretation for all states in all layers. Degrees below 4 connected to more nodes with such degrees finds the general house. Middle nodes are degree 3 in this structure; Top nodes are connected to both middle nodes; Bottom nodes are connected to one.

Let us now look at the BA-SHAPES dataset to classify each node with its position in a house motif, or if the node is part of the Barabasi-Albert base graph. Nodes in the base graph generally have a high degree, we can find house nodes as being connected to almost no high-degree nodes, but only among each other. From there we subdivide house nodes: The middle nodes have 3 house neighbors, the top node is connected to both the middle nodes, and the bottom nodes only to one.

| Layer | State | Decision Rule | Interpretation |
|---|---|---|---|
| Encoder | 0 | All nodes | No node features available for differentiation. |
| Layer 0 | 0 | Less than four state 0 neighbors | Degree 3 or lower nodes. (lower degree) |
| Layer 0 | 1 | At least four state 0 neighbors | Degree 4 or higher nodes. (high degree) |
| Layer 1 | 2 | At most 3 neighbors of state 1 | At most three high-degree neighbors. (House candidates, all nodes in the house have lower degree, the basegraph has many high-degree nodes). |
| Layer 1 | 4 | At least four neighbors in state 1 | At least four high-degree neighbors. |
| Layer 2 | 0 | Three state 2 neighbors | Three house candidates in the neighborhood (middle nodes in the house). |
| Layer 2 | 1 | At most state 2 neighbors | Top or bottom of the house. |
| Layer 2 | 4 | Previous state 4 nodes | At least four high-degree neighbors. |
| Layer 3 | 0 | At most one state 3 neighbor | Nodes connected to zero or one middle house node. |
| Layer 3 | 3 | Previous state 0 | Middle house nodes. |
| Layer 3 | 4 | At least two state 0 neighbors | Top house nodes. |
| Layer 4 | 0 | At least one state 3 neighbor and previous state 0 | Bottom house node: nodes connected to one house middle node. |
| Layer 4 | 1 | No state 3 neighbors | Nodes not connected to the house. |
| Layer 4 | 2 | At least one state 3 neighbor but not previous state 0 | Top node in the house. |
| Decoder | Not in house | Layer 2 state 4 | At least four high-degree neighbors. |
| Decoder | Middle of House | Not above and layer 3 state 3 | Three house candidates in the neighborhood. |
| Decoder | Bottom of House | Not above and layer 4 state 0 | Nodes connected to one house middle node. |
| Decoder | Top of House | Otherwise (but GraphChef could have used layer 3 state 4) | Otherwise (or nodes connected to two house middle nodes). |

Table 9: Analysis of the GraphChef recipe in Figure A.5 for the BA-Shapes dataset. The base graph has many edges, house nodes stand out by having a degree of 3 or lower and being connected to such nodes.

## B GRAPHCHEF ON DATASETS WITH MANY INPUT FEATURES

| Dataset | Features | GIN | GraphChef | | |
|---|---|---|---|---|---|
| | | | Differentiable | No pruning | Lossless pruning |
| CORA | 1433 | 0.87±0.02 | 0.82±0.03 | 0.69±0.04 | 0.68±0.03 |
| CiteSeer | 3703 | 0.77±0.01 | 0.70±0.03 | 0.61±0.04 | 0.61±0.02 |
| PubMed | 500 | 0.88±0.01 | 0.87±0.01 | 0.85±0.01 | 0.85±0.01 |
| OBGN-Arxiv | 128 | 0.68±0.02* | 0.68±0.01 | 0.28±0.11 | - |

Table 10: GraphChef results for citation datasets with high-degree counts. *Since the dataset has 40 classes, we use a state-size of 50 for GraphChef variants and 128 wide embeddings for GIN.

In the following, we want to discuss GraphChef on high-dimensional datasets such as Cora (1433) features. Table 10 shows a comparison of GIN, dish GNN and GraphChef similar to Table 1a. The results are mixed: on Pubmed, GraphChef performs comparable to GIN, on Cora there is a small drop for dish GNN but a significant drop when converting to trees. For CiteSeer, both dish GNN and converting to trees cause clear drops in accuracy. We see two factors that make this dataset challenging: Large feature spaces make it harder to reduce to a categorical state. For example, for the Cora dataset, the encoder needs to reduce from 1433 to 10 features. This effect increases in GraphChef when we limit the number of leaves: Having 100 decision leaves means that a tree can have 99 decision nodes and look at most at 99 features. But such trees are already impractical to interpret. We found that even after pruning, trees often contain long paths of depth 20 or more. The problems aggravate on the larger OBGN-Arxiv dataset: dish GNN performs decently with a drop comparable to CiteSeer, but GraphChef drops drastically in accuracy. Furthermore, this dataset reveals the scalability limits for GraphChef's pruning method: Pruning requires the number of leaves squared many runs over the dataset and does not scale to this dataset.

Therefore, we believe that handling such datasets requires a different approach. In future work, we imagine that these issues could be addressed through approaches such as PCA, clustering, or special MLP construction techniques (Wu et al., 2017a; Schaaf et al., 2019) to reduce the input space without breaking the interpretability chain before applying GraphChef.

## C  GENERATING EXPLANATIONS

In this section, we describe in detail how we can use GraphChef recipes to derive importance scores for the classification of a single node/graph. As in many existing explanation methods, these scores form a heatmap over all nodes to identify important inputs.

Formally, we are going to compute scores of the form $\mathbb{R}^{N \times S \times N}$ where $N$ is the number of nodes and $S$ the number of categorical states. For simplicity, we assume that every layer has the same number of states. For one node $u$ and one state $s$ the explanation $e(u, s)$ is a real-valued vector that assigns every other node $v$ an importance how much $v$ contributes to $u$ being in state $s$. We accumulate the importance over layers.

The importance of every node $u$ for the encoder layer is initialized as $e(u, v) = \mathbb{1}_u$ for every state $v$, where $\mathbb{1}_u$ is a vector that is 1 at the index of $u$ and 0 everywhere else. In other words, every node is its own explanation after the encoder.

To compute the explanation update for node $u$ in a GraphChef layer, we investigate its decision tree. First, we compute the Tree-Shap values for $u$ in the decision tree. These values reveal how important each decision feature in the tree is for predicting $u$; a value of 0 corresponds to an unused decision. Depending on the type of decision feature — a state feature, a message feature, or a delta feature (see possible cases in Figure 3) — we will add explanation to nodes differently. We handle each decision feature independently and weigh it with its Tree-Shap value.

**State features.** There are $S$ possible state features that can each lead to $S$ different new states. This yields $S \times S$ Tree-Shap values that we denote with $\tau_S(s, s')$. To compute explanations, we additionally require the indicator variable $\text{sign}(s)$ that is 1 if $u$ is in state $s$ at the start of the layer, and $-1$ otherwise. This indicator allows us to measure negative evidence that $u$ is *not* in a certain state. The "propagation" of state features is then easy, since all importance stays with the node.

$$\sigma(u, s') = \sum_{s \in S} \tau_S(s, s') \cdot e(u, s) \cdot sign(s)$$

**Message features.** There are also $S$ message features that can lead to $S$ different states; thus, we have $S \times S$ Tree-Shap values $\tau_M(s, s')$. Computing the explanations for a neighbor feature gives importance to each neighbor in the state $s$, normalized by the number of neighbors. Let $N(s)$ denote $u$'s neighbors in state $s$:

$$\mu(u, s') = \sum_{s \in S} \tau_M(s, s') \cdot \sum_{v \in N(s)} \frac{e(v, s)}{|N(s)|}.$$

**Delta features.** We have $S^2 - S$ delta features where $(s, s')$ encodes the feature that there are more neighbors in state $s$ than neighbors in $s'$. Here, we use the Tree-Shap values $\tau_\Delta(s, s', s'')$. We also need the indicator variables $(\mathbb{1}_{>(s,s')})$ that are 1 if indeed more neighbors are in the state $s$ rather than $s'$ and 0 if not. Now, explanation for delta features is similar to that of neighborhood features, where the majority class contribution is positive and the minority class contribution is negative:

$$\delta(u, s'') = \sum_{s \in S} \sum_{s' \neq s \in S} \tau_\Delta(s, s', s'') \frac{\sum_{v \in N(s)} e(v, s) - \sum_{v \in N(s')} e(v, s')}{|N(s)| + |N(s')|} \cdot \mathbb{1}_{>(s,s')}.$$

These explanations are added to those of the previous layers:

$$e(u, s) = e(us, s) + \sigma(u, s) + \mu(u, s) + \delta(u, s).$$

**Decoder layer** The decoder layer is slightly special since it uses skip connections. For node classification, we directly concatenate all intermediate features and use the same computation scheme to compute the final explanations. For graph classification, we additionally need to pool the nodes. We do this layer-wise and supply the decoder layer with per-layer node counts per state. The decoder can then use counting and comparison features similar to $M$ and $\Delta$ features in the GraphChef layers. The only difference is that instead of propagating the explanation to neighbors, we now need to propagate it to all of the nodes in the graph that were in the corresponding states.

# D DATASETS

## D.1 SYNTHETIC DATASETS

- **Infection** Faber et al. (2021) is a synthetic node classification dataset. This dataset consists of randomly generated directed graphs, where each node can be healthy or infected. The classification task predicts the length of the shortest directed path from an infected node.

- **Negative Evidence** Faber et al. (2021) is a synthetic node classification dataset. A random graph is created with ten red nodes, ten blue nodes, and 1980 white nodes. The task is to determine whether the white nodes have more red or blue neighbors.

- **BA Shapes** Ying et al. (2019) is a synthetic node classification dataset. Each graph contains a Barabasi-Albert (BA) base graph and several house-like motifs attached to random nodes of the base graph. The node labels are determined by the node's position in the house motif or base graph.

- **Tree Cycle** Ying et al. (2019) is a synthetic node classification dataset. Each graph contains an 8-level balanced binary tree and a six-node cycle motif attached to random nodes of the tree. The classification task predicts whether the nodes are part of the motif or tree.

- **Tree Grid** Ying et al. (2019) is a synthetic node classification dataset. Each graph contains an 8-level balanced binary tree and a 3-by-3 grid motif attached to random nodes of the tree. The classification task predicts whether the nodes are part of the motif or the tree.

- **BA 2Motifs** Luo et al. (2020) is a synthetic graph classification dataset. Barabasi-Albert graphs are used as the base graph. Half of the graphs have a house-like motif attached to a random node, and the other half have a five-node cycle. The prediction task is to classify each graph on whether it contains a house or a cycle.

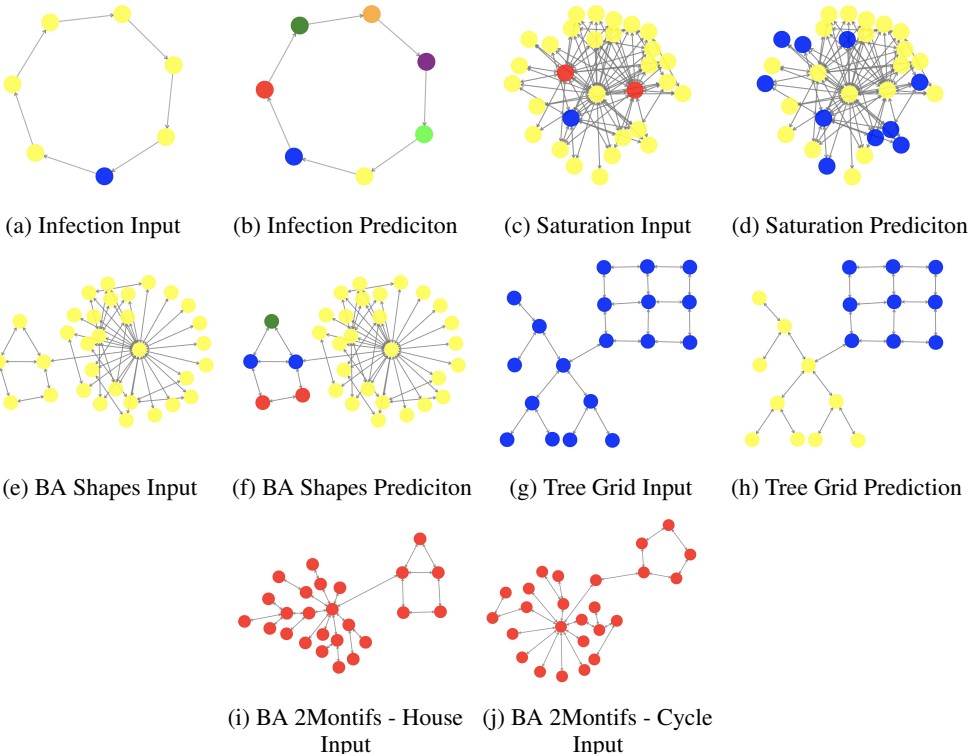

(a) Infection Input     (b) Infection Prediciton     (c) Saturation Input     (d) Saturation Prediciton

(e) BA Shapes Input     (f) BA Shapes Prediciton     (g) Tree Grid Input     (h) Tree Grid Prediction

(i) BA 2Montifs - House Input     (j) BA 2Montifs - Cycle Input

Figure 11: Synthetic Benchmarks - Example Graphs

| Dataset | Graphs | Classes | Avg. Nodes | Avg. Edges | Features |
|---|---|---|---|---|---|
| Infection | 1 | 7 | 1000 | 3973 | 2 |
| Negative Evidence | 1 | 2 | 2000 | 102394 | 3 |
| BA Shapes | 1 | 4 | 700 | 4110 | 0 |
| Tree Cycle | 1 | 2 | 871 | 1942 | 0 |
| Tree Grid | 1 | 2 | 1231 | 3130 | 0 |
| BA 2Motifs | 1000 | 2 | 25 | 50.96 | 0 |

Table 11: Statistics of Synthetic Datasets

## D.2 REAL-WORLD DATASETS

- **MUTAG** Debnath et al. (1991) is a molecule graph classification dataset. Each graph represents a nitroaromatic compound, and the goal is to predict its mutagenicity in Salmonella typhimurium. Mutagenicity is the ability of a compound to permanently change the genetic material, usually DNA, in an organism and therefore increase the frequency of mutations. The nodes in the graph represent atoms and are labeled by atom type. The edges represent bonds between atoms.

- **Mutagenicity** Kazius et al. (2005) is a molecular graph classification dataset. Each graph represents the chemical compound of a drug, and the goal is to predict its mutagenicity. The nodes in the graph represent atoms and are labeled by atom type. The edges represent bonds between atoms.

- **BBBP** Wu et al. (2017b) is a molecule graph classification dataset. Each graph represents the chemical compound of a drug and the goal is to predict its blood-brain barrier permeability. The nodes in the graph represent atoms and are labeled by atom type. The edges represent bonds between atoms.

- **PROTEINS** Borgwardt et al. (2005) is a protein graph classification dataset. Each graph represents a protein that is classified as an enzyme or not as an enzyme. The nodes represent the amino acids, and an edge connects two nodes if they are less than 6 angstroms apart.

- **REDDIT BINARY** Borgwardt et al. (2005) is a social graph classification dataset. Each graph represents the comment thread of a post on a subreddit. The nodes in the graph represent users, and there is an edge between users if one responded to at least one of the other's comments. A graph is labeled according to whether it belongs to a question-/answer-based or a discussion-based subreddit.

- **IMDB BINARY** Borgwardt et al. (2005) is a social graph classification dataset. Each graph represents the ego network of an actor/actress. In each graph, the nodes represent actors/actresses, and there is an edge between them if they appear in the same film. A graph is labeled according to whether the actor/actress belongs to the Action or Romance genre.

- **COLLAB** Borgwardt et al. (2005) is a social graph classification dataset. A graph represents a researcher's ego network. The researcher and his collaborators are nodes, and an edge indicates collaboration between two researchers. A graph is labeled according to whether the researcher belongs to the field of high-energy physics, condensed matter physics, or astrophysics.

- **Cora**, **CiteSeer**, and **PubMed** are popular citation networks (Yang et al., 2018b). Nodes are papers, and citations are edges. Nodes contain features that represent words of their contents and are labeled by subfields.

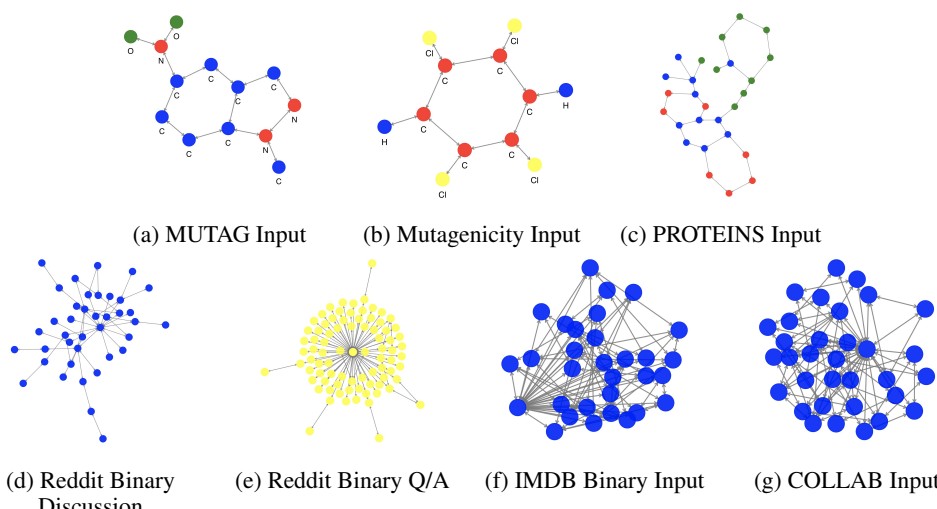

(a) MUTAG Input  (b) Mutagenicity Input  (c) PROTEINS Input

(d) Reddit Binary Discussion  (e) Reddit Binary Q/A  (f) IMDB Binary Input  (g) COLLAB Input

Figure 12: Real-world benchmarks - Example graphs

| Dataset | Graphs | Classes | Avg. Nodes | Avg. Edges | Features |
|---|---|---|---|---|---|
| MUTAG | 188 | 2 | 17.93 | 39.59 | 7 |
| Mutagenicity | 4337 | 2 | 30.32 | 61.54 | 14 |
| BBBP | 2039 | 2 | 24.06 | 51.91 | 9 |
| PROTEINS | 1113 | 2 | 39.06 | 145.63 | 3 |
| REDDIT BINARY | 2000 | 2 | 429.63 | 995.51 | 0 |
| IMDB BINARY | 1000 | 2 | 19.77 | 193.06 | 0 |
| COLLAB | 5000 | 3 | 74.49 | 4914.43 | 0 |
| Cora | 1 | 7 | 2485 | 5069 | 1433 |
| CiteSeer | 1 | 6 | 2110 | 2668 | 3703 |
| PubMed | 1 | 3 | 19717 | 44324 | 500 |

Table 12: Statistics of Real-World Datasets

# E  EXPERIMENT SETUP

We do a $10-$fold cross-validation of the data with different splits and train both GraphChef and a baseline GIN architecture. GNNs are trained on the training set for 1500 epochs, allowing early stopping on the validation loss with patience of 100. Each split uses early stopping on the validation score. Both GNNs use a $2-$ layer MLP for the update function, with batch normalization (Ioffe and Szegedy, 2015) and ReLu (Nair and Hinton, 2010) in between the two linear layers. We use 5 layers of graph convolution. GIN uses a hidden dimension of 16, GraphChef uses a state space of 10. We also further divide the training set for GraphChef to keep a holdout set for pruning decision trees. After we train dish GNN with gradient descent, we distill the MLPs into decision trees. Each tree is limited to having a maximum of 100 nodes.

GraphChef allows us to self-tune some hyperparameters. After training, we can inspect how many layers and categorical states are actually used in the recipe for a dataset. Table 13 shows the layers and states used for each dataset. After training, we can inspect the recipes from GraphChef and validate if all states and layers are necessary. If we find that the recipe uses fewer layers or states, we retrain with that number of layers or states. Table 13 shows the number of layers and states that we find per dataset. A full model is used for GIN.

| Layers | States | |
|---|---|---|
| Infection | 5 | 6 |
| Negative | 1 | 3 |
| BA-Shapes | 5 | 5 |
| Tree-Cycles | 5 | 5 |
| Tree-Grid | 5 | 5 |
| BA-2Motifs | 4 | 6 |
| MUTAG | 4 | 6 |
| Mutagenicity | 3 | 8 |
| BBBP | 3 | 5 |
| PROTEINS | 3 | 5 |
| IMDB-B | 3 | 5 |
| REDDIT-B | 2 | 5 |
| COLLAB | 3 | 8 |

Table 13: Tuned hyperparameters through GraphChef self-inspection

## F  USING THE TOOL

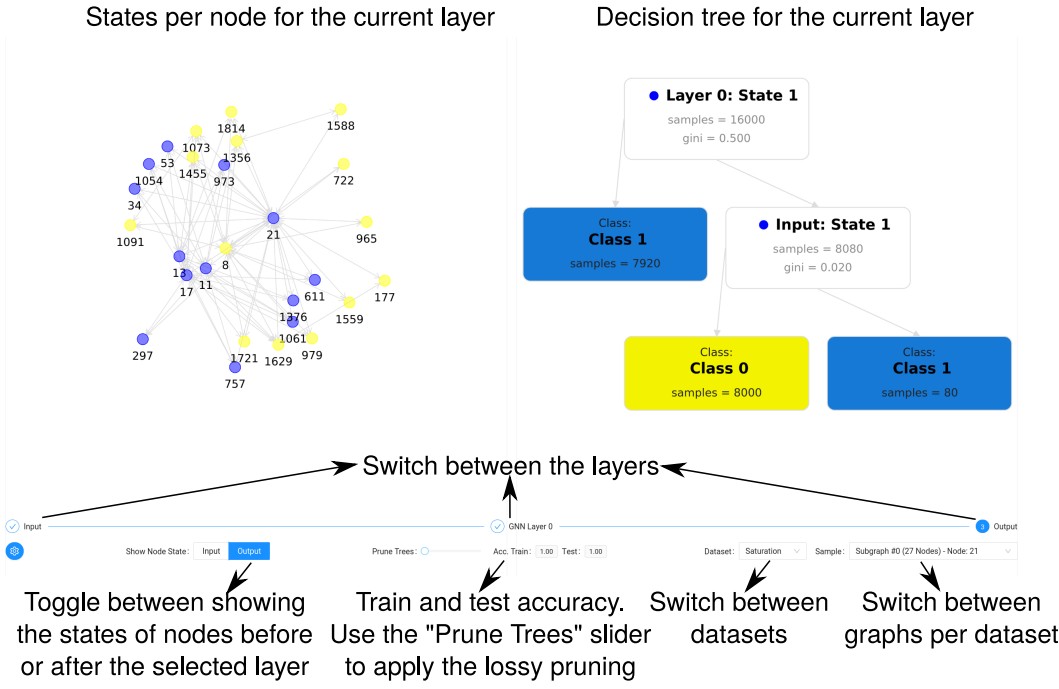

Figure 13: Initial page for the web tool. We can see the decision trees for GraphChef per dataset and which node for a graph is in what state. We can switch layers, graphs, and datasets. We can also see the test accuracy for the current setting and choose an amount of lossy pruning. with the slider.

An example instance of the tool is deployed and available via Netlify[2] and can be accessed under the link `https://interpretable-gnn.netlify.app/`. The supplementary material also contains code to host the interface yourself, in case you want to try variations of GraphChef. In the backend, we use PyTorch (Paszke et al., 2019)[3] and PyTorch Geometric (Fey and Lenssen, 2019)[4] to train GraphChef and SKLearn(Pedregosa et al., 2011)[5] to train the decision trees.

The tool is built with React, in particular the Ant Design library.[6] We visualize graphs with the Graphin library.[7] The interface is a single page that will look similar to Figure 13.

The largest part of the interface is taken up by two different panels at the top. In the right panel, you can see the decision tree for the currently selected layer. The trees use the three branching options from Figure 3. In the interface, evaluating the branching to true means taking the left path (this is opposite to Figure 3, which we will flip). In the left panel, you can see an example graph and which nodes end up in which state after this layer (on the bottom left you can toggle to see the input states instead). This panel does not show the full graph (most graphs in the datasets are prohibitively large), but rather an excerpt around an interesting region. Directly below these two graphics, you have the option to switch between layers by clicking on the respective bubble.

In the bottom right, you can switch to a different graph in the same dataset or to a different dataset. In the center, you can see the accuracy of GraphChef with the displayed layers. The slider allows one to apply the lossy pruning from Section 3.3 and the accuracy values update to the selected pruning level.

---

[2]`https://netlify.com/`

[3]`https://github.com/pytorch/pytorch`

[4]`https://github.com/pyg-team/pytorch_geometric`

[5]`https://github.com/scikit-learn/scikit-learn`

[6]`https://github.com/ant-design/ant-design/`

[7]`https://github.com/antvis/Graphin`

The interface also allows us to examine a single node more closely by clicking on it (see Figure 14; here we clicked on the blue node on the very right). Selecting reveals two things: In the graph panel, you can see the explanation scores from Section C for this node in this layer. In the tree panel, you can see the decision path in the tree for this node. This is particularly helpful if multiple leaves in the tree would lead to the same output state as in this example.

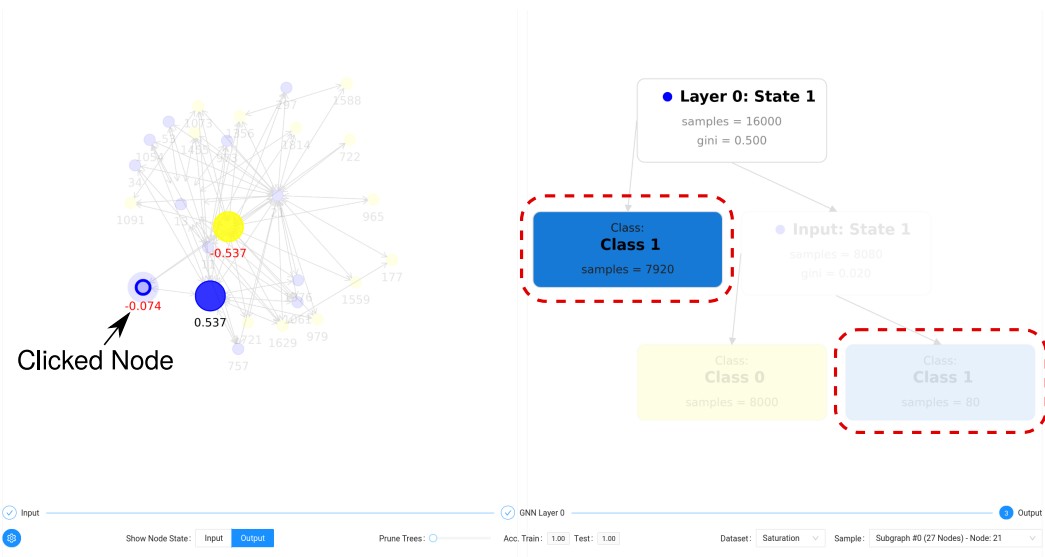

Figure 14: Interface when clicking on a node for closer examination. We can see node-level importance scores for this node on the left and the decision path taken on the right. Two paths end in the blue state, shown by the red boxes. The path the node takes is highlighted, the other path is blurred out.

