# OpenReview forum: "GraphChef: Decision-Tree Recipes to Explain Graph Neural Networks"
_ICLR.cc/2024/Conference — ICLR 2024 poster_

### Official Review · Reviewer_2W2r · 2023-10-20

**Soundness:** 3 good
**Presentation:** 3 good
**Contribution:** 3 good
**Rating:** 6
**Confidence:** 4

**Summary:**

The described work presents GraphChef, a method for achieving explainability in Graph Neural Networks (GNNs). It distills GNNs into decision trees with categorical hidden states using Gumbel softmax, essentially creating interpretable decision trees from the trained GNN. The decision trees are then pruned for interpretability based on accuracy. Evaluations on benchmark datasets show that while GraphChef maintains accuracy on some tasks, it may sacrifice performance on others in favor of obtaining an interpretable decision tree.

**Strengths:**

1. The article presents an innovative approach to achieving interpretability in Graph Neural Networks (GNNs) by replacing original network modules with interpretable ones, demonstrating that this approach maintains comparable learning performance. This is a novel solution to a significant problem in GNN research.

2. The paper differentiates itself from previous work by focusing on the extraction of global decision rules for entire datasets rather than providing localized explanations for individual outputs. This unique perspective on GNN interpretability is practical and valuable.

3. The article is well-structured, easy to understand, and supported by comprehensive empirical evaluations. It also offers a user-friendly web interface for visualizing decision trees, making it accessible for downstream usage. This approach is both novel and promising for improving the interpretability of GNNs.

**Weaknesses:**

1. The method's applicability is restricted to graphs with very few features, making it unsuitable for more complex datasets. The authors acknowledge this limitation, but addressing it could involve techniques like dimension reduction or prototype learning when discretizing the hidden states to handle more feature-rich graphs.

2. The proposed method's use of one-hot hidden states may reduce the model's expressiveness, especially for complex problems. For these, the resulting decision trees could become overly complex and challenging to interpret.

3. The method's performance in terms of explanation is described as only "comparable" to post-hoc methods, and it may not offer significant advantages over using standard GNN models followed by post-hoc interpretability techniques.

**Questions:**

How does the model perform on large datasets?

---

> ### Author Response · Authors · 2023-11-18
>
> Thank you for your review and your comments and comments. We hope that our answers below help resolve any questions or potential misunderstandings.
>
> > The method's performance in terms of explanation is described as only "comparable" to post-hoc methods, and it may not offer significant advantages over using standard GNN models followed by post-hoc interpretability techniques.
>
> We think there has been a misunderstanding. The models are comparable if we focus on the explanation scores in Table 2. But we think GraphChef offers a superior form of explanation as we show in Figure 1. We see the main improvement and also the main contribution of this paper is that we can not only find the important nodes/edges but also explain *why* thes nodes/edges are important. Note that standard post-hoc methods only give you a heatmap of importances, which constitutes a fundamental loss of information on why exactly the model made the given choices. So it’s impossible to recover true, not some implied or supposed, decision making logic from these heatmaps. Which is a fundamental limitation of the standard post-hoc methods.
>
> > How does the model perform on large datasets?
>
> Larger datasets may also pose problems for interpretability: Generally, larger datasets come with more edge cases which require more states and larger trees to adequately handle. Trees can become harder to interpret. Two approaches to tackle this are using the lossy pruning to aggressively first understand the core predictions paths and slowly grow the trees. Furthermore, GraphChef also offers explanation scores like existing explanation methods that scale to large trees with many states. explanation scores that we compute like existing explanation methods scale with the number of states and layers and can help guide the user like existing methods do.

---

> > ### Comment · Reviewer_2W2r · 2023-11-21
> > **Response to authors**
> >
> > Dear authors,
> >
> > Thanks for the further clarification.  Could you also provide some comments on my first two weaknesses?

---

> > > ### Author Response · Authors · 2023-11-22
> > >
> > > Dear Reviewer:
> > >
> > > About your other two weaknesses:
> > >
> > > "restricted to graphs with very few features"
> > >
> > > It is true that our current paper focuses on graph aspects, and not so much on inputs where the graph aspect is in the background and the features of the nodes/edges are more important. We believe that our method could be interesting even if the graph aspect is completely in the background. For example, you might have feature-rich data that is just organized as a matrix. Then our method still might give you insights about how neighboring cells of the matrix interact. But since the prime focus of our paper is graphs, we didn't go there. Clearly there is a trade-off between expressiveness and learnability here, and your suggestions might make such matrix data understandable (and explainable) on a new (higher) level than just marking the important cells of the matrix.
> > >
> > > "one-hot hidden states"
> > >
> > > This is also a very good point, and we thought about this a lot when doing the research. Again it's a trade-off between learnability and expressiveness. We experimented considerably while setting up our system, and what we have now is what worked best. For instance, first we only had the pure one-hot states, but then we noticed that it might be very useful to compare the counts of two states (while still being learnable), so this is now in our system. But the possibilities are endless of course. You might say that real values should be added, or loops/recursion, since they will make the resulting explanations more useful (and simpler for humans). We fully agree that this would be an exciting direction. We didn't go there because so far we couldn't handle the learning part. What we have now is a good compromise to explain the reason while still being learnable.
> > >
> > > Your two comments can also be combined, and then it gets really interesting. We believe that our general approach of explainability is interesting well beyond graphs. Let's take for instance the image domain: A future version of our approach might be able to explain MNIST, but give you an algorithm (not just a decision tree but really a little algo description) why an image is a certain number. In natural language sort of like "This is a 9 because it has a circle on top and a vertical stroke attached to that circle at the bottom right, where by circle I mean ... and by stroke I mean ..." somewhat explained as little algorithms. Something as useful as LLMs but with precise algorithms.
> > >
> > > In other words, we believe that what we can do for graphs will (eventually) be interesting for ML in general, and give a new way of explaining what is going on.

---

> > > > ### Comment · Reviewer_2W2r · 2023-11-22
> > > > **Follow up**
> > > >
> > > > Thanks for the authors' reply.  I raised my score.

---

### Official Review · Reviewer_8CuG · 2023-10-30

**Soundness:** 3 good
**Presentation:** 3 good
**Contribution:** 3 good
**Rating:** 8
**Confidence:** 2

**Summary:**

This paper introduces GraphChef, a Graph Neural Network (GNN) model that integrates decision trees to provide human-comprehensible explanations for each class in a dataset. The authors note that while GNNs are popular for graph-based domains, they are often black-box models that lack interpretability. GraphChef aims to address this issue by generating decision trees that show how different features contribute to each class. The authors demonstrate the effectiveness of GraphChef on the PROTEINS dataset and other explanation benchmarks that require graph reasoning. They also highlight the importance of small trees to ensure that the generated recipes are understandable to humans. Overall, GraphChef provides a promising approach for generating interpretable explanations for GNNs.

**Strengths:**

Originality:
- The integration of decision trees with GNNs to provide human-comprehensible explanations is a novel approach that has not been explored extensively in the literature.
- The authors also introduce a new benchmark dataset for graph-based explanation methods, which can be used to evaluate the effectiveness of different models.

Quality:
- The authors provide a thorough evaluation of GraphChef on multiple datasets and benchmarks, demonstrating its effectiveness in generating interpretable explanations for GNNs.
- The paper includes detailed discussions of the limitations and future work of GraphChef, which can guide future research in this area.

Clarity:
- The paper is well-organized and clearly written, making it easy for readers to follow the authors' arguments and understand the technical details of GraphChef.
- The authors provide several examples and visualizations to illustrate the effectiveness of GraphChef in generating human-comprehensible explanations.

Significance:
- The lack of interpretability of GNNs is a significant challenge in many graph-based domains, and GraphChef provides a promising approach for addressing this issue.
- The authors highlight the potential applications of GraphChef in safety-critical domains such as medicine, where it is important to understand how decisions are made.
- The paper also contributes to the broader goal of developing explainable AI methods, which can increase trust and transparency in machine learning models.

**Weaknesses:**

The authors note that GraphChef can struggle to create recipes for datasets with a large feature space, which limits its applicability in some domains. Future work could investigate alternative methods for combining multiple features into one split to address this limitation.

**Questions:**

I have no question.

---

> ### Author Response · Authors · 2023-11-18
>
> Thank you for your review and your comments. We agree with you that the novel decision-tree powered explanations are a promising approach for interpretability and that tackling the datasets with large feature spaces is the a very promising direction for future work.

---

### Official Review · Reviewer_VFKG · 2023-11-01

**Soundness:** 3 good
**Presentation:** 3 good
**Contribution:** 3 good
**Rating:** 6
**Confidence:** 3

**Summary:**

The paper presents an interesting idea of learning decision trees to get explainable models for graph learning tasks. Tree models have a long history and are well known for their explainability. Direct training of tree models on a graph is not straightforward. This work proposes a method of casting a decision tree from a trained neural model. In the empirical evaluation, the tree model shows slight accuracy drop but provides much better explanations.

**Strengths:**

The idea of training decision trees from graph data is novel. It distills neural models to decision trees. It has several designs (e.g. via the dish network) that make the distilling possible.

The results of the paper show that the proposed method can provide better explanations while has slight performance drop.

**Weaknesses:**

In the empirical evaluation the work shows good performance. However, there should be some theoretical bound over the ability of the tree model. For example, it cannot exceed the expressiveness of the 1-WL algorithm. At the same time, it is unknown whether it can even match the ability of the 1-WL algorithm. I think a theoretical analysis is missing from the work.

**Questions:**

A few key steps are missing from the paper. Here are two questions.

First, how does the model pool the information to get a graph representation?

Second, the dish model uses "Gumbel-Softmax" to GNN layers. As far as I know "Gumbel-Softmax" is designed for sampling continuous variables that approximate the categorical distribution. I don't understand why randomness is needed here.

Third, "We bootstrap the internal state h^0_v with an encoder layer on the initial node features x_v" -- what does "bootstrap" mean?

Fourth, what does "encoder" and "decoder" mean? Do you use an autoencoder to recover the input or not?

Fifth, how many layers do you use in your model?

---

> ### Author Response · Authors · 2023-11-18
>
> Thank you for your review and your comments and comments. We hope that our answers below help resolve any questions or potential misunderstandins.
>
>
> > I think a theoretical analysis is missing from the work.
>
> We made a maybe too short theoretical argument at the end of the Subsection 3.1. In principle, instead of having a $log(d)$ bits, we can use $d$ many encoded bits. For example, we can represent a GIN that uses just a single embedding that we assume is encoded in 32 bit floats as a stone age GNN with 2^32 = states (which is roughly 4 million). This is of course impractical and impossible to train, but, in principle, the stone-age layers also have 1WL power.
>
> If we only concern ourselves with the graph structure (ignore node features for a moment), one can easily see that actually just O(n) states are enough to achieve 1-WL expressive power, as the original 1-WL algorithm that GIN is able to simulate uses discrete states and at most needs n different colors (states). So theoretically it is straightforward to make GraphChef as expressive as standard GNNs (1-WL). But since the GNN expressive power hierarchy is not fine grained it's hard to characterize precisely the impact of shrinking the categorical space has on expressive power. That is why we focused on the empirical comparison to GIN.
>
> > First, how does the model pool the information to get a graph representation?
>
> Let us look at an example graph with two layers and three states. The output layer would get pooled information per layer: How many nodes are in state 1, 2, 3 after the first layer and how many nodes are in state 1, 2, 3. after the second layer. In total, there would be 6 features available to the output layer. Of course, the number of states must not be the same per layer.
>
> > Second, the dish model uses "Gumbel-Softmax" to GNN layers. As far as I know "Gumbel-Softmax" is designed for sampling continuous variables that approximate the categorical distribution. I don't understand why randomness is needed here.
>
> That is correct, the randomness is only used during training to allow exploring a wider decision space. Especially in the beginning of training, the model predictions are purely random and sampling allows us to explore the search space well. We increase the softmax temperature during training to converge the distribution towards a one-hot distribution. Removing the randomness at the start of the training might result in gradient descent getting stuck. During testing we turn the Gumbel-Softmax off completely in favor of a direct argmax.
>
> > Third, "We bootstrap the internal state h^0_v with an encoder layer on the initial node features x_v" -- what does "bootstrap" mean?
> > Fourth, what does "encoder" and "decoder" mean? Do you use an autoencoder to recover the input or not?
>
> We used the terms encoder and decoder to emphasize the layers that do not have any GNN message passing. These GNN layers start with categorical values on each node and produce a categorical value on each node. On the other hand, the encoder translates the input features from the input domain (which need not be categorical, for example if we perform regression) to categorical states. The decoder (or readout) layer reads the intermediate and final states (potentially pooled) as outlined above to compute a prediction.
>
> “Bootstrap” was probably a slightly misleading term to use here. We will clarify this to something similar to the above.
>
> > Fifth, how many layers do you use in your model?
> We allow the model to use up to five layers. Afterwards, we inspected the recipes how many layers the model actually use - Table 13 in Appendix E shows how many layers were actually used.

---

> > ### Comment · Reviewer_VFKG · 2023-11-22
> > **Thank you for your clarification**
> >
> > Thank you for your clarification

---

### Official Review · Reviewer_2TRZ · 2023-11-04

**Soundness:** 3 good
**Presentation:** 4 excellent
**Contribution:** 3 good
**Rating:** 8
**Confidence:** 4

**Summary:**

This paper proposes an explainable GNN model by combining decision trees into message passing framework. The idea is interesting and novel. The paper is well written and easy to understand. The main advantage of the proposed method is its ability to explain the whole dataset as compared to the existing methods that focus on explaining individual graphs in the data.

**Strengths:**

The main strength is its ability to explain the whole dataset as compared to the existing methods that focus on explaining individual graphs in the data.

**Weaknesses:**

n/a

**Questions:**

1. Please clarify the usage of Gumbel-Softmax to the latent embeddings in detail. I had a hard time understanding this.

2. Why is performance of GIN and GraphChef is similar as listed in Table 1?

---

> ### Author Response · Authors · 2023-11-18
>
> Thank you for your review and your comments and comments. We hope that our answers below help resolve any questions or potential misunderstandins.
>
> > Please clarify the usage of Gumbel-Softmax to the latent embeddings in detail. I had a hard time understanding this.
> The Gumbel-Softmax happens at the very end of the embedding computation. In principle, you could imagine that we first run a normal GIN to produce embeddings which are $d$ numbers. Now we additionally apply the Gumbel Softmax on those numbers to instead sample a one-hot encoded vector of size $d$. Now this has the following implications:
>
> We already do this to the inputs before the first message passing layer (and every subsequent one). Therefore, the node embeddings (which are the messages) for all of the message passing layers are already one-hot encoded. The summation of neighborhood messages becomes counting the neighbors in each state.
> When we want to replace the neural functions by a decision tree, we can treat each layer like a tabular classification problem: In the beginning of each layer we have three inputs: the previous one-hot state, the message counts, and the pairwise differences. We want to predict the output of the layer, which is a categorical variable.
>
> > Why is performance of GIN and GraphChef is similar as listed in Table 1?
> The selling point of Table 1 is precisely that we can have a fully interpretable model that performs similarly to standard GNNs (i.e. GIN).
> Note that GraphChef has the same expressive power as GIN if sufficiently many categorical states are used ( O(n) ).  To have an easily interpretable model we do use smaller embedding spaces, but the similar performance of GIN and GraphChef hints that a very high information throughput in the messages (high-dimensional embeddings) is not necessary for these tasks in particular.

---

### Official Review · Reviewer_U2So · 2023-11-06

**Soundness:** 3 good
**Presentation:** 2 fair
**Contribution:** 3 good
**Rating:** 6
**Confidence:** 4

**Summary:**

This paper integrates decision trees into the message-passing framework of GNNs and proposes a self-explanatory GNN model called GraphChef. Inspired by the Stone-age model, GraphChef utilizes the Gumbel-Softmax function to induce categorical latent states for GNN layers, then uses the categorical inputs and outputs and distills them into decision trees. Since the decision-making process of decision trees is comprehensible to humans, GraphChef is able to provide a series of recipes to help us understand the behavior of GNNs. The authors have designed a series of pruning strategies for training decision trees to prevent overfitting, and experiments show that while providing interpretability, GraphChef also ensures expressive power on par with GIN on certain datasets.

**Strengths:**

- Unlike previous approaches that identify substructures or sub-features of the original input most relevant to the output, GraphChef offers a unique perspective for interpreting GNNs. The "recipes" it returns reflect higher-level behaviors of the model. Concurrently, GraphChef still provides a method to calculate heatmap-style importance scores for individual graphs.
- The authors have designed a diverse and comprehensive set of network architectures for GraphChef, ensuring its suitability for both graph classification and node classification tasks.
- The authors have developed pruning strategies for GraphChef that enhance readability and prevent overfitting, while only sacrificing a minimal amount of performance.
- The recipes extracted by GraphChef on some small datasets are in alignment with human intuition.

**Weaknesses:**

- As the authors mentioned, a limited number of categorical states may restrict its expressive and interpretive power on large datasets, since the feature space of the inputs in large datasets is often vast. It is conceivable that the decision trees generated by GraphChef would also be immense, and an overly large decision-making process can be difficult for humans to directly comprehend.
- For self-explanatory models, we usually need to understand the trade-off between interpretability and expressive power. However, I did not see such an analysis in this paper, which means we are unable to comprehend the model's balance between expressiveness and interpretability.
- Pairwise Comparisons require O(n^2) space complexity, which is impractical for large-scale datasets (if a large state size is needed).
- Although GraphChef can provide a series of decision rules to help us understand the behavior of GNNs at a higher level, the authors' experiments are all based on small-scale datasets (most of which are synthetic), and for complex real-world datasets, these decision rules may still be complex and difficult for humans to understand, especially when the decision trees are deep.
- While the introduction of decision trees into the interpretation of GNNs is novel, concepts such as the Stone-age model, transforming categorical states to decision trees, and decision tree pruning are not new.

**Questions:**

- Regarding the training process of the decision trees, how are the three different types of branches (as shown in Figure 3) chosen during the training procedure? Also, which training algorithm is employed, C5.0 or CART?

- Why does the structure of GraphChef often require more than five layers?

- Why does the expressive power of GraphChef surpass GIN on certain datasets, and can you provide some qualitative analysis?

---

> ### Author Response · Authors · 2023-11-18
>
> Thank you for your review and your comments and comments. We hope that our answers below help resolve any questions or potential misunderstandins.
>
> > However, I did not see such an analysis in this paper, which means we are unable to comprehend the model's balance between expressiveness and interpretability.
>
> Theoretically, the differentiable categorical layer is actually as expressive as a standard GNN (e.g. GIN) if sufficiently many states are used. The original 1-WL test that upperbounds the standard GNN expressive power as proposed originally actually uses categorical states (colors) and needs up to N of them. As GNN expressive power hierarchy is not very fine grained, besides the k-WL levels, it’s hard to analyze the gradual expressive power loss when reducing the number of categorical states below N. However it is worth noting that in very many cases one can get away with running the 1-WL algorithm successfully with the number of colors << N.
>
> Practically, we do not have to heavily pre-constrain the model to a particular tradeoff of expressiveness and interpretability. Rather, we ask the decision trees to maximize expressiveness with a weak constraint for interpretability: Every decision tree is limited to 100 nodes (for computational efficiency). We leave the balance to the human after trees are trained through pruning: By choosing which level of deterioration is acceptable we can trade-off large but accurate trees versus smaller and more interpretable ones. This dynamic pruning can also be easily seen and experimented with in our provided web interface.
>
> > Regarding the training process of the decision trees, how are the three different types of branches (as shown in Figure 3) chosen during the training procedure? Also, which training algorithm is employed, C5.0 or CART?
>
> The branches are chosen by the decision tree learning algorithm. Let us assume the state size of 3 as in Figure2: This means that decision trees receive a 12 long input vector to predict the categorical output. By the way we order the 12 features: the first 3 features correspond to the one-hot encoded previous state and we use a branch like Figure 3a). The next 3 features correspond to the message aggregation with a branch like Figure 3b) and the remaining features are delta features as in Figure 3c).
>
> We used the sklearn version of CART https://scikit-learn.org/stable/modules/tree.html#tree-algorithms-id3-c4-5-c5-0-and-cart
>
> > Why does the structure of GraphChef often require more than five layers?
>
> GraphChef actually uses five or less layers: Table 13 in the Appendix shows the number of layers per dataset. We trained an initial version with 5 layers for all dataset, but inspecting the recipes shows that GraphChef does not always use all 5 available layers.
>
> > Why does the expressive power of GraphChef surpass GIN on certain datasets, and can you provide some qualitative analysis?
>
> Does this question refer to the results in Table 1 for the datasets Infection, BA-Shapes, BBBP, PROTEINS, or REDDIT-B? The scores are very close and overlapping when we consider the standard deviations. We attributed these to chance. The perhaps only substantial outperformance is by GIN on the Mutagenicity dataset.

---

> > ### Comment · Reviewer_U2So · 2023-11-22
> > **Acknowledgement**
> >
> > Thank you for the reply.

---

### Meta-Review · Area_Chair_8kfK · 2023-12-06

**Metareview:**

The authors develop a self-explanatory GNN by integrating decision trees into the message passing. The explanations demonstrate how different features contribute to the datasets' classes in a way that is aligned with human intuition.

The reviewers find the approach novel and very interesting. As reveiwers U2So puts it, the approach offers a unique perspective on GNN interpretation and, indeed, the recipes it generates reflect higher-level behaviors of the model.

There were several reviewer comments related to insufficient analyses or explanations, however the authors have largely addressed those comments in the rebuttal.

The method is not applicable for datasets with large feature spaces, but the authors have been very open about this during the rebuttal phase, and I thank them. They do mention that larger datasets come with more edge cases hinting towards a tradeoff for interpretability, and although this is just an intuition it does somehow help with this concern.

Overall, despite its limitations, the paper is expected to be of interest to the community and open up new avenues for research.

**Justification For Why Not Higher Score:**

Several limitations still exist, notably the method not being applicable to large data.

**Justification For Why Not Lower Score:**

This paper offers a unique perspective, complementing previous methods on explanability for graph tasks. Despite its limitations, I think it can open up new research avenues.

---

### Decision · Program_Chairs · 2024-01-16

Accept (poster)